# Stochastic variation in the initial phase of bacterial infection predicts the probability of survival in *D. melanogaster*

David Duneau[1,2]*, Jean-Baptiste Ferdy[2], Jonathan Revah[1,3], Hannah Kondolf[1‡], Gerardo A Ortiz[1], Brian P Lazzaro[1,3†], Nicolas Buchon[1,3†]*

[1]Department of Entomology, Cornell University, Ithaca, United States; [2] Laboratoire Évolution & Diversité Biologique, UMR5174 EDB, CNRS, ENSFEA, Université Toulouse 3 Paul Sabatier, Toulouse, France; [3]Cornell Institute of Host Microbe Interactions and Disease, Cornell University, Ithaca, United States

**\*For correspondence:**
david.duneau@gmail.com (DD);
nicolas.buchon@cornell.edu (NB)

[†]These authors contributed equally to this work

**Present address:** [‡]Case Western Reserve University School of Medicine, Cleveland, United States

**Competing interests:** The authors declare that no competing interests exist.

**Abstract** A central problem in infection biology is understanding why two individuals exposed to identical infections have different outcomes. We have developed an experimental model where genetically identical, co-housed *Drosophila* given identical systemic infections experience different outcomes, with some individuals succumbing to acute infection while others control the pathogen as an asymptomatic persistent infection. We found that differences in bacterial burden at the time of death did not explain the two outcomes of infection. Inter-individual variation in survival stems from variation in within-host bacterial growth, which is determined by the immune response. We developed a model that captures bacterial growth dynamics and identifies key factors that predict the infection outcome: the rate of bacterial proliferation and the time required for the host to establish an effective immunological control. Our results provide a framework for studying the individual host-pathogen parameters governing the progression of infection and lead ultimately to life or death.

DOI: https://doi.org/10.7554/eLife.28298.001

## Introduction

Two hosts exposed to apparently identical infections may have dramatically different outcomes. In one case, the host may experience only mild symptoms and recover easily, while in another the host may suffer devastating illness or even death. In many pathogenic infections, only a fraction of the infected hosts die from the infection or carry a high pathogen burden, while other hosts may strongly limit parasite proliferation. For example, mosquitoes infected with malaria experience high and low pathogen load (*Molina-Cruz et al., 2008*), rats infected with *Enterococcus faecalis* experience bimodal pathogen burden (*Frank et al., 2015*), and even the abundance of gut microbes exhibits bimodal variation (*Lahti et al., 2014*). Genetically identical hosts may still experience drastically different infection outcomes (e.g. clonal population of *Daphnia magna* infected by one strain of bacteria [*Duneau et al., 2012*]). These observations suggest that the process of infection could itself be intrinsically variable. While such differences can sometimes stem from genetic or environmental variation among hosts, in other cases the difference in outcome appears to be more arbitrary or stochastic, and the relative contributions of genetic, environmental and random individual variation remain unclear.

Variation in within-host disease processes can affect pathogen evolution by modulating the within-host evolutionary dynamic during the course of the infection (*Alizon et al., 2011*; *Mideo et al., 2011*). Theory predicts that variation in traits such as bacterial growth rate or host response time to an infection can affect the outcome of the infection and thus the epidemiology of

**eLife digest** Sick individuals do not all respond to an infection in the same way. One individual may experience mild symptoms and recover easily, while another may suffer devastating illness or even death. A number of factors are often assumed to account for these differences, including the sex, age and genes of the individuals, and differences in the environments the individuals have been exposed to. However, random variations in how an individual's immune system interacts with the infection could also play an important role in recovery.

Duneau et al. have now studied how genetically identical fruit flies who were raised in the same environment respond to different bacterial infections. This enabled them to develop a mathematical model that describes how a bacterial infection develops in an individual. In an initial phase, the bacteria proliferate freely. If the immune defenses activate in time to control the infection, the number of bacteria in the fly decreases to a constant level and the infection enters a long-term, or chronic, phase. The sooner this happens the more likely it is that the fly will survive. If the immune control happens too late, the infection enters a terminal phase and the fly will die once the number of bacteria increases to a certain level.

The model therefore reveals that the precise time at which the immune system takes control of the bacterial population – termed the "Time to Control" – determines the outcome of the infection. Duneau et al. confirmed this by injecting bacteria into identical flies. A small variation in the Time to Control was sometimes the difference between a fly living or dying. Understanding what controls this apparently random variation is key to understanding infection and potentially developing more efficient treatments for a wide range of diseases – not just those caused by bacteria, but also those caused by viruses and parasites, like HIV and malaria.

DOI: https://doi.org/10.7554/eLife.28298.002

the disease (*Alizon et al., 2011*; *Fenton et al., 2006*). To better understand the selection pressures that shape host-parasite interactions, it is important to empirically open the 'within-host black box' and characterize the different phases of an infection and to define the evolutionarily relevant parameters in the host and parasite (*Alizon et al., 2013*).

*Drosophila* is a prime model to study host-microbe interactions (*Buchon et al., 2014*). *Drosophila* relies on both humoral and cellular responses to fight infection (*Lemaitre and Hoffmann, 2007*). The humoral response consists of the secretion of antimicrobial peptides (AMPs) into the hemolymph (insect blood) and the activation of the melanization cascade, both of which limit microbial proliferation. The cellular responses include encapsulation of foreign bodies and phagocytosis by dedicated immune cells. Antimicrobial peptides are constitutively expressed in a restricted set of barrier tissues at low levels but are massively upregulated in the fat body in response to infection. Two major signaling pathways, the Toll and Imd pathways, drive the induction of AMPs in the fat body. Importantly, we and other groups have observed strong inter-individual variation in the outcome of *Drosophila* infected with a wide range of bacteria and viruses (*Chambers et al., 2014*; *Clemmons et al., 2015*; *Ferreira et al., 2014*; *Howick and Lazzaro, 2014*; *Kutzer and Armitage, 2016*), which suggests that *Drosophila* is a good model to study the stochastic nature of infection.

In the present work, we aim to identify key parameters of the intra-host dynamic that underlie the inter-individual variation in the outcome of infection. We first demonstrate that infection of *Drosophila melanogaster* by a series of bacterial pathogens leads to highly variable infection outcomes. Notably, infections with the Gram-negative *Providencia rettgeri* and the Gram-positive *Enterococcus faecalis* result in one of two binary outcomes: some individuals die with a high pathogen burden, whereas others survive indefinitely with fairly asymptomatic and low-burden persistent infections. Different individuals of the same age and genotype may experience either outcome, even when cohoused in the same rearing vial. We find that bacterial burden upon host death did not correlate with the time post-injection at which death occurs, and that the lethal burden varies across bacterial species and strains. Surviving flies sustain a much lower pathogen burden, which also varies across bacterial species and can depend on the initial inoculum and on the immune response. Using a variety of experimental manipulations, we determined that inter-individual variation occurs as a consequence of the interplay between bacterial growth rate and the activation of the host immune

response. We developed a theoretical model that describes the probabilistic nature of bacterial growth and thus host survival based on three key factors: the net rate of bacterial proliferation ($\mu$), the timing of effective immunological control ($T^c$), and an inferred threshold pathogen density ($n^{tip}$) that must be reached to enable the bacteria to switch to a lethally acute infection instead of a chronically persistent one. Our model predicts infection dynamics accurately and suggests that inter-host variation in survival must originate in the ability of the pathogen to reach the $n^{tip}$ threshold before effective immune control is established. We observe empirically that the quantitative activation of the immune response varies among individual hosts, suggesting that inter-individual stochasticity early in infection might lead to differences in the probability of ultimate survival. Altogether, our results illustrate the mechanisms underlying the variable nature of infection outcome and provide a framework for studying the distinct parameters that govern the progression of infection and lead ultimately to life or death.

## Results

### The outcome of infection for individual hosts is highly variable

Infection with different pathogens produces dramatically different outcomes that range from the certain death of all host individuals to the nearly certain survival of all host individuals. In order to define the range of outcomes, we infected male *Drosophila melanogaster* with a panel of seven bacteria by injection of a controlled dose into the abdominal cavity (*Figure 1A*). Infection with *Providencia alcalifaciens* and *Serratia marcescens Db11* was lethal to the flies in a short timeframe, killing all individuals within one day of infection. Conversely, nearly all individuals survived infection with *E. coli* and *Erwinia carotovora Ecc15* (formally known as *Pectinobacterium carotovorum carotovorum 15*) (*Figure 1A*). Infections with different natural pathogens of *Drosophila* including *P. rettgeri*, *E. faecalis*, *P. burhodogranariea* resulted in the death of only a fraction of the infected population (*Figure 1A*). It is commonly assumed that inter-individual differences can stem from genetic differences or environmental variation, including nutrition, gut microbes or rearing conditions. We therefore controlled for these sources of variation by infecting same-aged, genetically identical hosts (isogenic lines of the Drosophila Genome Resource Panel and $w^{1118}$), reared in a common environment with a controlled microbiota and given experimentally identical infections (*Figure 1B*). Even under experimentally identical conditions, we continued to see dual outcomes of infection. In addition, the relative probability of the infection achieving either outcome varied among genotypes. Inter-individual variation could also stem from small variation in fly handling at the time of infection. For instance, differences in exposure to the $CO_2$ used for anesthesia during the experiments could influence the probability of death upon infection (*Helenius et al., 2009*). However, we found that even 15 min of exposure to $CO_2$ prior to or after injection did not affect the chance of survival relative to 2 min of exposure (*Figure 1—figure supplement 1*). These results demonstrate stochastic variation in the outcome of infection (life or death) even among genetically identical individuals, although host and bacterial genotype combine to determine the probability of surviving infection.

One could reasonably predict that individuals that survive their infections may do so by having cleared the infecting pathogen. To test this, we quantified bacterial loads in individual flies that were still alive seven days after injection with *E. coli*, *E. carotovora*, *E. faecalis*, and *P. rettgeri*. At this point, mortality curves had plateaued and little to no further death was observed (*Figure 1A*). We found that none of the bacteria had been entirely cleared from the host and nearly all individual hosts still had measurable loads of bacteria (*Figure 1C*). In the case of the most benign pathogens, which did not lead to the death of any host individuals, a few host individuals either cleared the infection or sustained loads below our detection threshold of 10 bacteria per fly (~22% for *E. coli* and ~8% for *E. carotovora*). However, most surviving hosts developed persistent, chronic infections. In addition, the bacterial load sustained during the persistent phase of infection was constant but distinct for each bacterium (e.g., *Figure 1—figure supplement 2A*). We termed this parameter the 'Set Point Bacterial Load' (*SPBL*), in analogy to the commonly used 'Set Point Viral Load' for chronic viral infections (*Fraser et al., 2014*). Interestingly the *SPBL* for *E. coli* and *E. carotovora* were significantly lower than those from *E. faecalis* and *P. rettgeri* infections, which also differed between each other (*Figure 1C*). The *SPBL* for *E. coli* and *E. carotovora*, our two least virulent bacteria, was a few hundred bacteria per infected host. In contrast, the *SPBL* for pathogens of intermediate virulence,

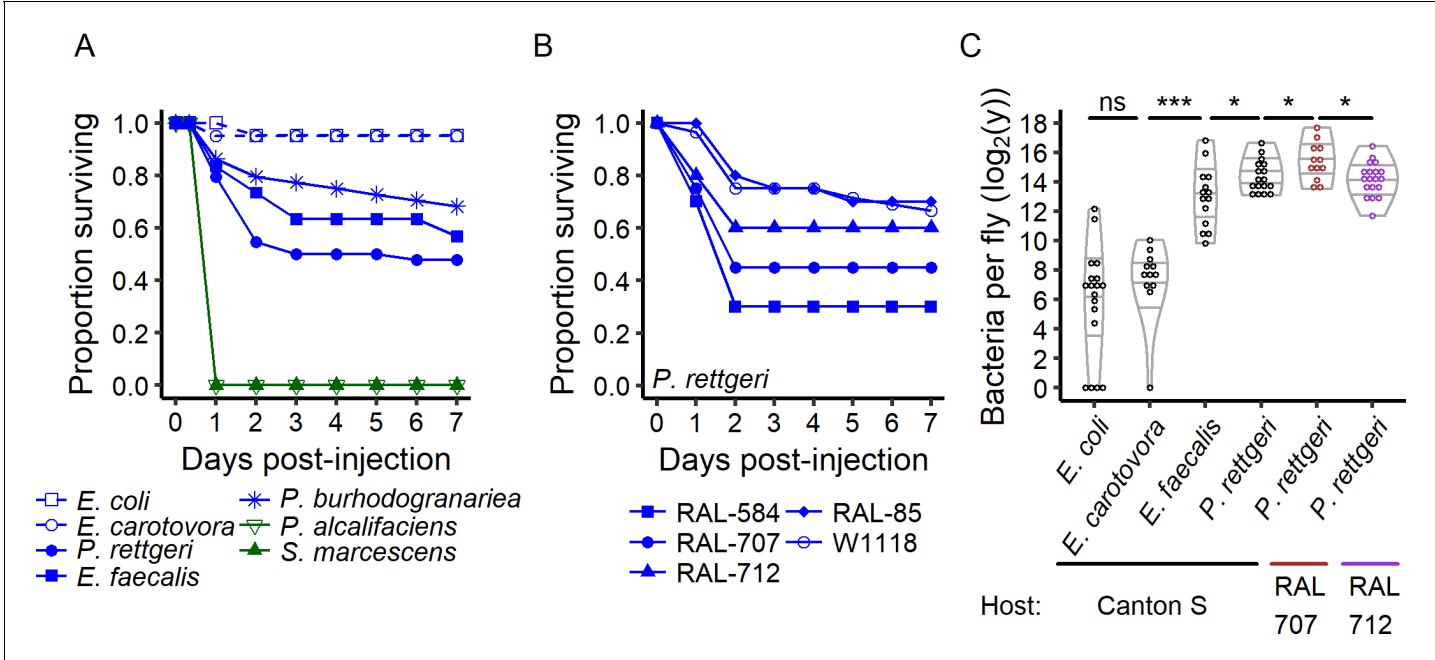

**Figure 1.** The outcome of infection ranges from 100% to 0% survival. (**A**) Canton S flies were injected with the same inoculum ($OD_{600}$ = 1, ca. 30,000 bacteria) of different bacteria species. *Providencia alcalifaciens* (n = 29) and *Serratia marcescens* (n = 30) are lethal pathogens (green solid lines) and killed 100% of the hosts in less than a day. *E. coli* (n = 21) and *E. carotovora* (n = 20) infection did not reduce host survival (blue dashed lines). *P. rettgeri* (n = 44), *P. burhodogranariea* (n = 44) and *E. faecalis* (n = 30) infections led to intermediate outcomes (blue solid lines): some individuals died in the first two days after infection while others survived. (**B**) Even for males of the isogenic lines of the *Drosophila* Genome Resource Panel and of w[1118], same age and reared in a common environment with a controlled microbiota, only a fraction of the hosts was killed by experimentally identical infections with *P. rettgeri*. This fraction depended on the host genotype (Coxph: Line: df = 4, $\chi^2$ = 19.23, p=0.0007, $n_{RAL-584}$ = 20, $n_{RAL-85}$ = 20, $n_{RAL-707}$ = 20, $n_{RAL-712}$ = 20, $n_{w1118}$ = 84). (**C**) Flies that survive infection are chronically infected with a constant bacterial load we termed Set-Point Bacterial Load (*SPBL*). Distinct bacterial species differ in their *SPBL* (Kruskal-Wallis test: df = 3, $\chi^2$ = 63.14, p=1.25e-13). Host genotype also impacts *SPBL* (Kruskal-Wallis test: df = 2, $\chi^2$ = 6.63, p=0.03). The annotations above the violin plots reflect results of two-by-two Wilcoxon tests comparing medians: *p<0.05, ****p<0.0001 and ns: p>0.05.

DOI: https://doi.org/10.7554/eLife.28298.003

The following source data and figure supplements are available for figure 1:

**Source data 1.** Data set for *Figure 1*.
DOI: https://doi.org/10.7554/eLife.28298.006
**Source data 2.** Data set for *Figure 1—figure supplement 1*.
DOI: https://doi.org/10.7554/eLife.28298.007
**Source data 3.** Data set for *Figure 1—figure supplement 2*.
DOI: https://doi.org/10.7554/eLife.28298.008
**Figure supplement 1.** The variability of outcome in infection is not due to variation in the time of exposure to $CO_2$ before and after injection.
DOI: https://doi.org/10.7554/eLife.28298.004
**Figure supplement 2.** The chronic phase of infection is characterised by the Set Point Bacterial Load (SPBL).
DOI: https://doi.org/10.7554/eLife.28298.005

such as *P. rettgeri* and *E. faecalis*, centered around a few thousand bacteria, and no single fly ever cleared the infection. In addition, we found the *SPBL* for *P. rettgeri* to be significantly different among three *D. melanogaster* host genotypes (*Figure 1C*), increasing with the size of the initial inoculum (*Figure 1—figure supplement 2B*).

We next examined the individual hosts that died from their infections. In the case of *P. rettgeri*, only a subset of infected hosts die. Mortality begins as soon as 16 hr post-injection and continues over the course of a few days before survivorship plateaus (*Figure 1A,B*). We quantified the Bacterial Load Upon Death (*BLUD*) by sampling individual flies within 30 min after their death, before the confounding effect of *post-mortem* bacterial proliferation. We compared those loads to pathogen burdens of still-living individuals sampled from the population all along the course of infection

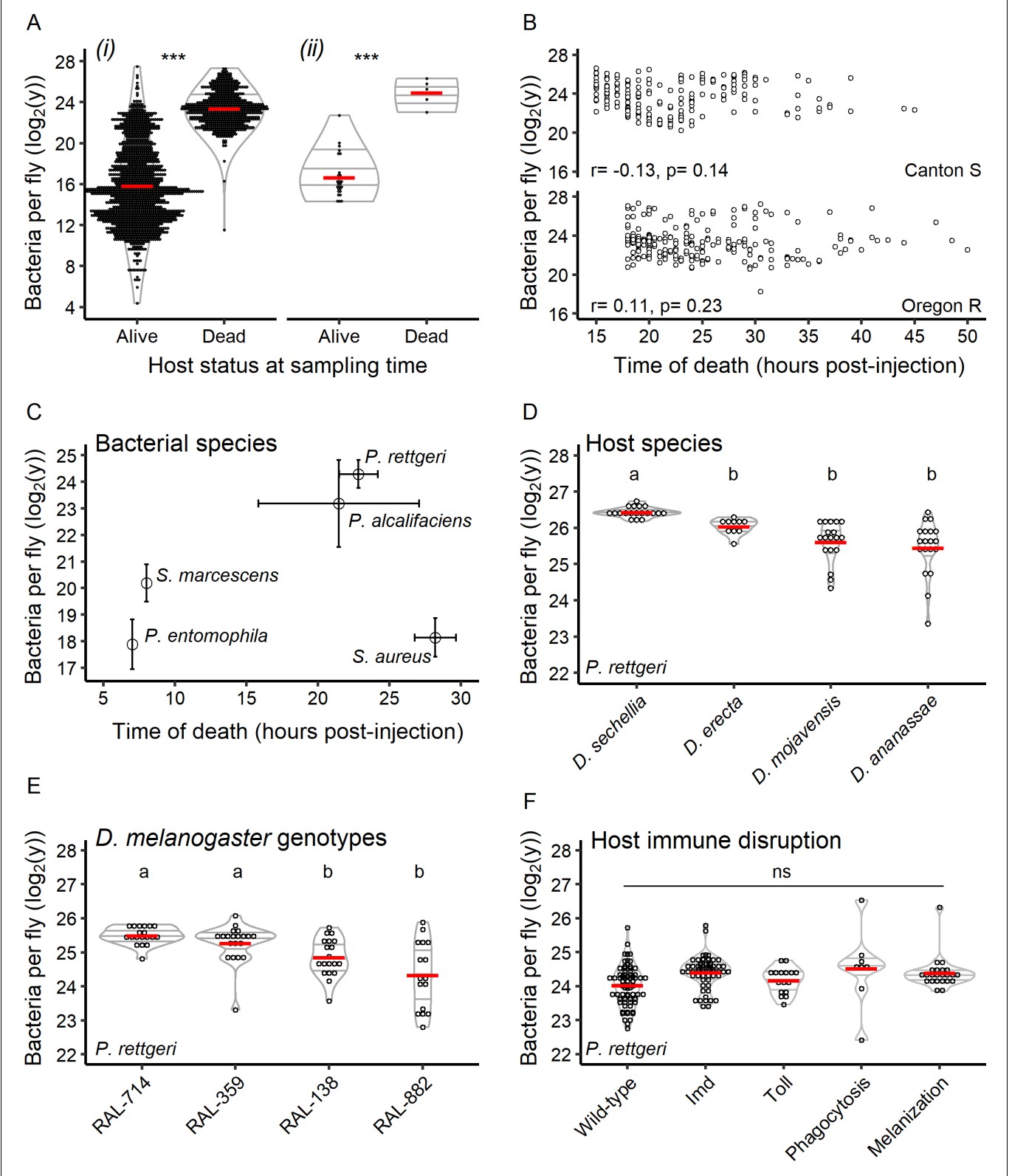

**Figure 2.** Hosts die at a set bacterial burden, the *BLUD*, which varies with host and pathogen. (A) The bacterial load in flies within 30 min of death from *P. rettgeri* infection is a constant and is higher than that of flies that did not die from the infection. This is true when comparing dead hosts to living hosts regardless of duration of infection (*i*: df = 1, W = 963036, p=1.39e-252) and when dead individuals are compared to living individuals sampled simultaneously (*ii*: df = 1, W = 140, p=0.0004). The stars above the violin plots reflect results of two-by-two Wilcoxon tests comparing medians:

*Figure 2 continued*

***$p<0.001$. (B) For both Canton S (Spearman correlation test - S = 462763.5, r = −0.13, p=0.14) and Oregon R (Spearman correlation test - S = 327036.4, r = 0.11, p=0.23), the time of host death did not correlate with the *BLUD* for *P. rettgeri*. (C) Different bacterial species can differ in *BLUD* when infecting the same host genotype (Kruskal-Wallis test: df = 3, $\chi^2$ = 92.14, p=3.45e-21). However, the average time to death for a given bacterial species does not correlate with its average *BLUD* (Spearman correlation test: S = 12, r = 0.4, p-value=0.51). (D) The *BLUD* of *P. rettgeri* depends on host species (Kruskal-Wallis test: df = 3, $\chi^2$ = 39.59, p=1.29e-08). (E) The *BLUD* of *P. rettgeri* depends on host genotypes (Kruskal-Wallis test: df = 3, $\chi^2$ = 23.438, p=3.27e-05). (F) The *BLUD* of *P. rettgeri* is not altered by removing components of the host immune system listed on the x-axis (linear mixed model with lines as random factor with df = 4, Deviance = 7.24, p=0.12). Wildtype refers to the wildtype lines Canton S, Oregon R and w[1118]. 'Imd' refers to the double mutant loss-of-function *PGRP-LC, PGRP-LE* and the loss-of-function mutant *Relish*. 'Toll' refers to the *spz*[rm7] mutant. Phagocytosis refers to the progeny of the cross *Hml-Gal4 >UAS* bax. 'Melanization' refers to a line double mutant for two prophenoloxidase genes (*PPO1Δ, PPO2Δ*). The annotations above each violin plot represent the result of the post hoc Dunn tests; treatments with the same letter are not significantly different after FDR p-value correction. Error bars in C are standard deviation around the mean.
DOI: https://doi.org/10.7554/eLife.28298.009

The following source data and figure supplement are available for figure 2:

**Source data 1.** Data set for *Figure 2*.
DOI: https://doi.org/10.7554/eLife.28298.011
**Source data 2.** Data set for *Figure 2—figure supplement 1*.
DOI: https://doi.org/10.7554/eLife.28298.012
**Figure supplement 1.** The BLUD is constant over time and does not depend on the initial dose.
DOI: https://doi.org/10.7554/eLife.28298.010

---

(*Figure 2Ai*) or at a unique time point where 5% of the population died at once (*Figure 2Aii*). The *P. rettgeri* loads in recently-dead individuals were $2^{23.4} \pm 2^{1.6}$ (mean ±sd) bacteria/fly, regardless of the time post-injection at which the host died (*Figure 2B* and *Figure 2—figure supplement 1A*). These loads were consistently 100-fold higher than those of living individuals ($2^{15.8} \pm 2^{3.8}$ bacteria/fly; *Figure 2A*). The narrow range of bacterial load upon death and the fact that live flies carry fewer bacteria than recently dead flies (*Figure 2Aii*) suggests that the *BLUD* is a lethal load and cannot be sustained in living individuals. We found that there is a *BLUD* associated with every bacterium and that the *BLUD* differed significantly between bacteria (*Figure 2C*), indicating that this threshold is a property of the pathogen. Surprisingly, the *BLUD* did not reflect the time it took various pathogens to kill the host. For example, the *BLUD* for *S. marcescens*, which killed wild-type flies within 8 hr of injection, was $2^{20.2} \pm 2^{0.7}$ bacteria/fly. This is lower than that of *P. alcalifaciens* ($2^{24.3} \pm 2^{0.5}$), which killed within 24 hr, but higher than for *S. aureus* ($2^{18.1} \pm 2^{0.7}$), which killed within 28 hr (*Figure 2C*). Because the *BLUD* varies across bacteria but is constant for a given bacterial strain in a given environment, we propose that it provides a novel measure of bacterial pathogenicity or toxicity to the host.

Reciprocally, we reasoned that the *BLUD* also represents a measure of host disease tolerance corresponding to the maximal bacterial load that can be sustained before the individual dies from the infection. As a measure of disease tolerance, the *BLUD* might vary among hosts. We therefore next measured both inter- and intra-specific variation in the *BLUD* after infection with *P. rettgeri*. We first assessed the *BLUD* of *P. rettgeri*-infected flies belonging to four different *Drosophila* species (*D. sechellia, D. erecta, D. mojavensis* and *D. ananassae*) and to four different *D. melanogaster* wild-derived isogenic lines (RAL-714, RAL-359, RAL-138 and RAL-882). We found that the *BLUD* varied significantly across *Drosophila* species (*Figure 2D*) and across different *D. melanogaster* genotypes (*Figure 2E*). To our surprise, however, the *BLUD* did not depend on the initial inoculum starting the infection (*Figure 2—figure supplement 1B*). We tested this by challenging wild-type *D. melanogaster* lines with *P. rettgeri* infection at five different doses ranging from ~300 ($OD_{600} = 0.01$) to ~150,000 ($OD_{600} = 5$) bacteria injected. Flies injected with higher doses died earlier than flies injected with lower doses (*Figure 2—figure supplement 1C–D*) but *BLUD* did not correlate with the dose injected (*Figure 2—figure supplement 1B*). These data affirm that hosts die at a lethal burden that does not depend on their initial infection dose or on their time to death.

Having described the BLUD as a measure of disease tolerance, we sought to determine whether the immune response, the main mechanism of bacterial elimination (i.e. resistance), affected the *BLUD*. We did not detect significant differences in the *BLUD* between any of the immune-deficient hosts and their corresponding wild-type genotypes (*Figure 2F*). This result supports the idea that

the BLUD is an exclusive measure of tolerance and not resistance. Collectively, our results indicate that bacterial infections of *Drosophila* result in binary outcomes. Either the host dies from acute infection with a stereotypical bacterial burden that is independent of dose, the *BLUD*, or the host survives but sustains a chronic infection at low and also stereotypical bacterial burden that does depend on initial dose, the *SPBL*. These binary outcomes are general to *D. melanogaster* and are observed in all wild-type host genotypes and species that we tested, although the relative proportion of hosts entering either state may vary with pathogen identity and host genotype (*Figure 1A and B*). Although the *BLUD* is a key parameter to describe bacterial virulence and host disease tolerance, it does not explain the observed inter-individual variation in the outcome of infection (death versus chronic persistence).

## The outcome of infection is predicted by inter-individual variation in within-host bacterial proliferation

We have shown that individual hosts may either survive or die from their infections, and we hypothesized that among-individual variation in survival might result from differences in bacterial growth dynamics at early stages of infection. In order to test this hypothesis, we initiated an in-depth study of bacterial growth in response to three categories of pathogens: those that quickly kill all infected hosts, those that kill none, and those that kill only a fraction of the infected population. We measured the bacterial load in individual flies at regular one-hour intervals for the first 16 hr after injection (*Figure 3*). Upon infection with two avirulent bacteria, *E. coli* and *E. carotovora*, bacterial burdens inside individual flies decreased soon after injection (*Figure 3A*), suggesting that these bacteria are controlled within a couple of hours by the host, even though they are not completely cleared. After injection with bacteria that kill all infected hosts in few hours (*P. alcalifaciens* or *S. marcescens*), the pathogen burdens increased monotonically until all individuals were dead (*Figure 3B*), suggesting that the host fails to control the infection. The pattern was very different for pathogens killing only a fraction of the infected host population: *E. faecalis*, *P. burhodogranariea*, *P. rettgeri* (*Figure 3C–E*). For these, we found that bacteria proliferated homogenously in all individuals during the early phase of infection (up to around 6–8 hr), but that subsequently two groups of hosts could be distinguished. Hosts of one group carried high and increasing bacterial loads, and we hypothesized that these individuals are fated to die. Hosts in the other group exhibited a smaller bacterial load that stabilized over time, and we hypothesize that these are the individuals who will sustain chronic infections. Remarkably, the in vivo proliferation of *P. rettgeri* in the early phase of infection is equivalent to the rate of in vitro growth in LB medium (*Figure 3—figure supplement 1*), suggesting that the bacteria are growing largely without constraint in this early phase. The same is true of the always-lethal *P. alcalifaciens* and *S. marcescens* (*Figure 3—figure supplement 1*). Proliferation continues at this same rate in the host group with the higher bacterial load up until the *BLUD* is reached, at which point the host dies. These results are consistent with the hypothesis that host variation in the outcome of infection might stem from differences in the ability of the fly to control infection in the early stage.

Although perhaps unlikely, one possible explanation for the observed data would be that a subset of hosts infected with *P. rettgeri*, *E. faecalis*, *P. burhodogranariea* do not survive their infections because a mutation that provides resistance to the host immune system occurs in the bacterial population, allowing these bacteria to continue to grow instead of being controlled. If this were true, the bacteria that kill the fly would have evolved hyper-virulence and be able to outcompete their wild-type counterparts. Conversely, the flies that survive their infections might conceivably do so because the bacteria infecting them sustained loss-of-virulence mutations. To test these hypotheses, we sampled bacteria from flies bearing either low (i.e. close to the *SPBL*) or high bacterial load (i.e. close to the *BLUD*; *Figure 4Ai*) at 12 hr post-injection and used them to infect new cohorts of flies. In both cases, we found that bacteria isolated from infected hosts were as likely or more likely than the original bacterial clone to kill new hosts, although we see no difference in mortality upon infection with bacteria from high-load versus low-load flies (*Figure 4Aii*). These data collectively provide no evidence that the probability of a host controlling or failing to control bacterial infection results from heritable changes in the bacterial population during infection.

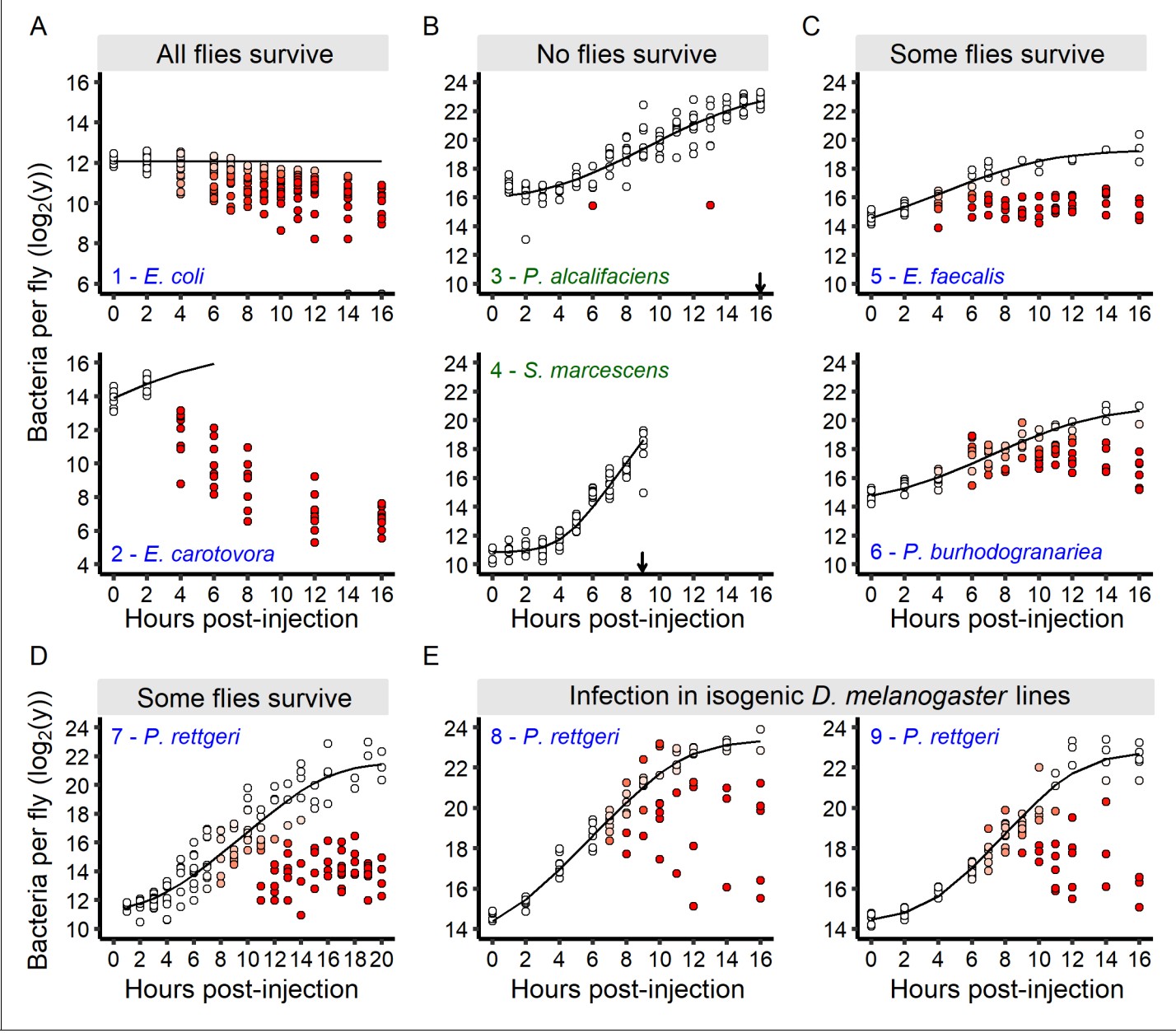

**Figure 3.** The infection outcome depends on inter-individual variation in within-host bacterial proliferation. (A) For bacteria that do not kill any infected hosts, bacterial loads decreased soon after the beginning of the infection. Grey dots on the x-axis represent individuals without detectable bacteria. (B) For bacteria that quickly kill all infected hosts, bacterial loads continuously increased until all hosts were dead (black arrows). (C-D) For bacteria that kill a fraction of infected individuals, an initial period of bacterial growth in each individual was followed by a period where individual flies diverged in their loads. In these infections, two groups of individual hosts appeared. In the first group, bacteria continue to grow, while in the other, bacterial growth was controlled. (E) The increase in inter-individual variation in bacterial load over time is also observed in fully isogenic populations (RAL-707 and RAL-712). In all panels, each dot represents the bacterial load in a single fly, the solid line represent the standard Baranyi bacterial population growth fitted on the white dots (see Materials and methods). The intensity of red in the dots represents the probability that hosts controlled the infection (i.e. the denser the red is within a dot, the higher is the probability that this host controlled the infection), and whose pathogen burdens are better described by an exponential decrease model (see Materials and methods). Measurements of bacterial load dynamics were taken from Canton S flies unless otherwise stated. The numbers in each panel identify each infection and are subsequently used in *Figure 6B and C*. Green labels are bacteria that kill all the hosts and blue labels are bacteria that establish persistent infection in surviving hosts.

DOI: https://doi.org/10.7554/eLife.28298.013

The following source data and figure supplement are available for figure 3:

**Source data 1.** Data set for *Figure 3*.

*Figure 3 continued on next page*

*Figure 3 continued*

DOI: https://doi.org/10.7554/eLife.28298.015

**Source data 2.** Data set for *Figure 3—figure supplement 1*.

DOI: https://doi.org/10.7554/eLife.28298.016

**Figure supplement 1.** In vivo proliferation in the early phase of infection is equivalent to the rate of in vitro growth in LB medium.

DOI: https://doi.org/10.7554/eLife.28298.014

## The host immune response determines the seemingly stochastic switch in bacterial growth

We next hypothesized that the host immune response might be primarily responsible for regulating pathogen burden, and that variation among individual hosts in immune activity might determine whether infections proceed to acute lethality or settle into chronic persistence. To test the hypothesis that the existence of a binary outcome to infection depends on the immune system, we studied *P. rettgeri* growth during an in vivo infection and the probability of death in hosts deficient for either the cellular or humoral immune responses.

Engulfment by macrophage-like cells, called phagocytes, is considered the first line of immune defense once the parasite is inside an insect. We thus tested whether phagocyte activity predicts inter-individual variation in bacterial growth. We infected hosts in which phagocytes were genetically ablated with *P. rettgeri* and compared them to infected control hosts (*Figure 5A*). Individual hosts separated into high-load and low-load groups over the first 16 hr of infection both in the presence and in the absence of phagocytes. We therefore concluded that phagocytosis is not a determinant of the switch between lethal and persistent *P. rettgeri* infection under our experimental conditions.

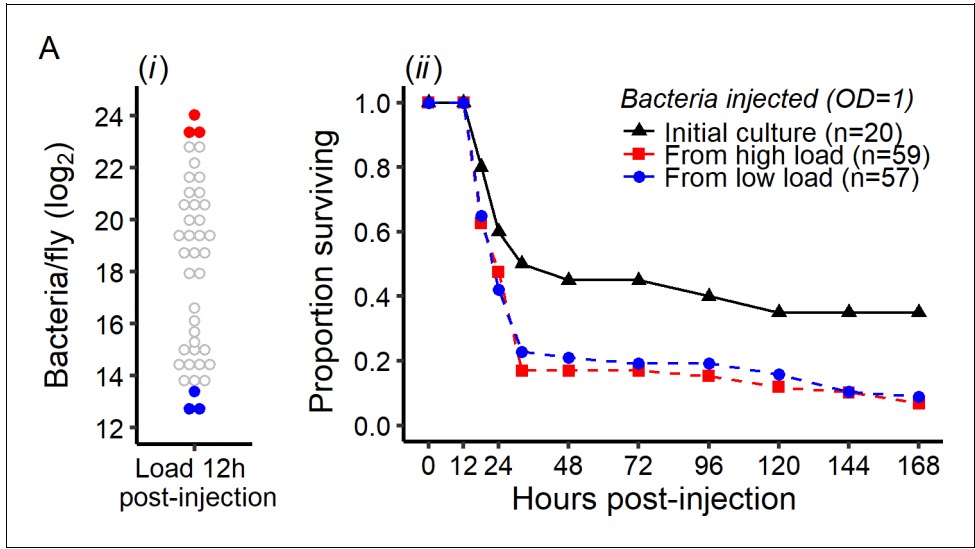

**Figure 4.** Early intra-host bacterial evolution does not explain the dual outcome of infection. Bacterial evolution inside the fly does not explain the binary outcomes of infection. (**A**) Quantification of bacterial load to determine bacterial populations killing the host (bacterial load close to the *BLUD* 12 hr post-injection, red dots) and bacterial population controlled by the host (bacterial load close to the *SPBL* 12 hr post-injection, blue dots), (**B**) Survival of Canton S flies injected with *P. rettgeri* from the stock used for the original injections, or recovered from flies in which bacteria had been growing for 12 hr (see panel A). Bacteria recovered from experimental flies (dashed lines) killed more hosts than the stock culture (solid line) (Coxph: df = 3, $\chi^2$ = 8.28, p=0.01). However, there were no differences in mortality caused by bacterial populations isolated from flies with high loads versus those isolated from flies that had controlled bacterial growth (Coxph: df = 2, $\chi^2$ = 0.07, p=0.78).

DOI: https://doi.org/10.7554/eLife.28298.017

The following source data is available for figure 4:

**Source data 1.** Data set for *Figure 4*.

DOI: https://doi.org/10.7554/eLife.28298.018

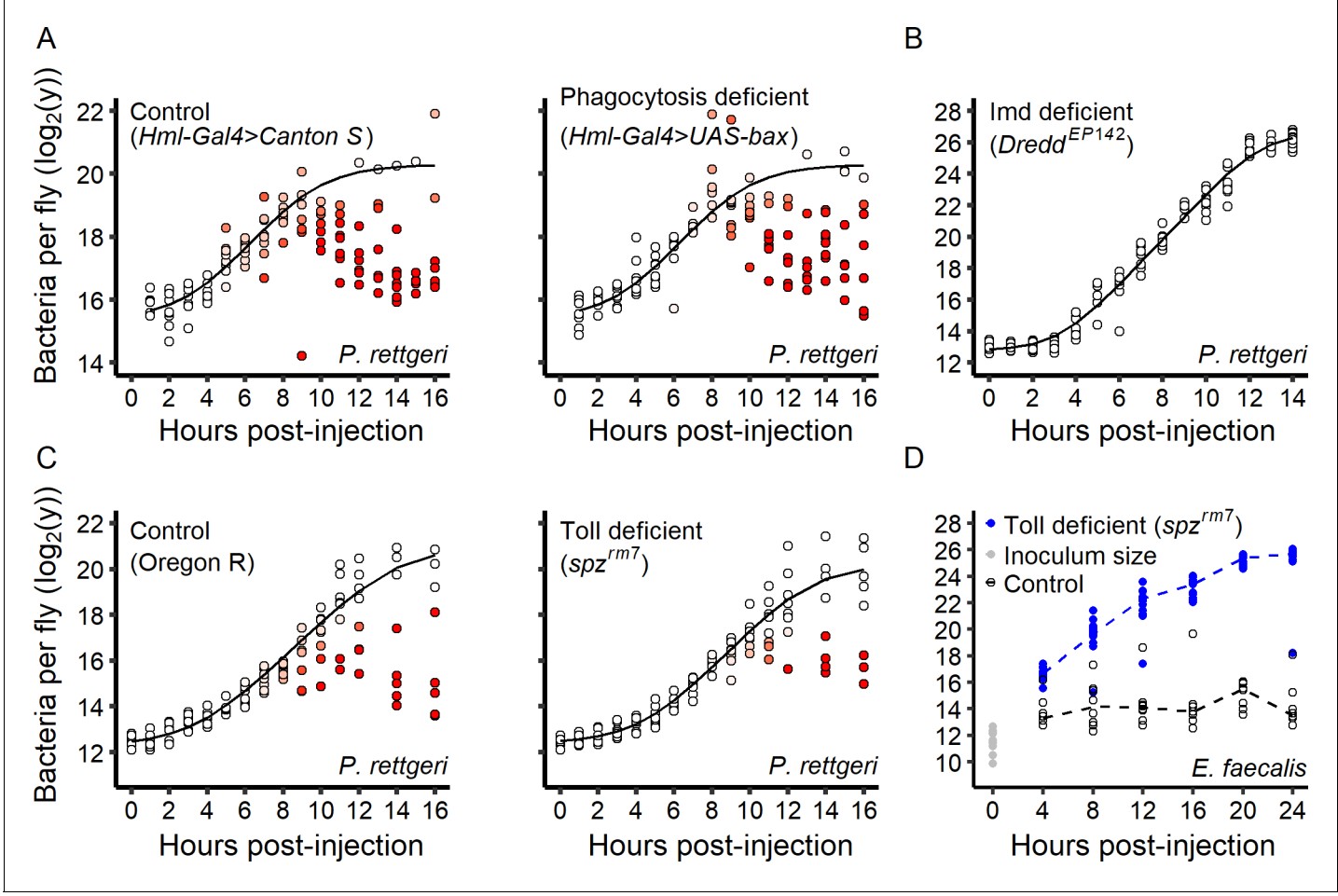

**Figure 5.** Role of the immune system in the infection dynamic. (**A**) Within-host *P. rettgeri* loads at different times post-injection for wild-type (Canton S) flies and flies deficient in phagocytosis (*Hml-Gal4 >UAS GFP; UAS-Bax, Gal80ts*). Distinct groups of high-load and low-load flies appear even in absence of phagocytosis. (**B**) All hosts infected by *P. rettgeri* suffer high pathogen load in the absence of an Imd-mediated immune response. In these hosts (*Dredd^EP142* mutants), *P. rettgeri* growth occurs at a rate similar to in vitro bacterial growth (modeled by a Baranyi model, black line). (**C**) We monitored within-host *P. rettgeri* loads at different times post-injection for Toll deficient (*spz^rm7* mutants) and wild-type flies (Oregon R). Distinct groups of high-load and low-load flies appear even in absence of the Toll-dependent immune response. (**D**) We monitored within-host *E. faecalis* loads at different times post-injection for Toll-deficient (*spz^rm7* mutants) and wild-type flies (Oregon R). For this Gram-positive bacterium, despite receiving the same initial inoculum (grey dots), bacterial load differed between mutant and wild-type flies by four hours post-injection (Welsh t-test, df = 22.57, t = 13.01, p-value=5.6e-12). In absence of the Toll pathway (blue dots), all hosts suffered high pathogen burdens. In all panels, each dot represents the bacterial load in a single fly. In Panels A-C, the solid line represent the standard Baranyi bacterial population growth fitted on the white dots (see Materials and methods). The intensity of red in the dots represents the probability that hosts controlled the infection, and whose pathogen burdens are better described by an exponential decrease model (see Materials and methods).

DOI: https://doi.org/10.7554/eLife.28298.019

The following source data is available for figure 5:

**Source data 1.** Data set for *Figure 5*.

DOI: https://doi.org/10.7554/eLife.28298.020

We next investigated the contribution of the humoral response to inter-individual variation in bacterial control. Gram-negative bacteria are mainly controlled by the Imd pathway, a major regulator of AMP production (*Buchon et al., 2014*). We tested whether flies deficient for the Imd pathway (*Dredd^EP142* mutant, *Figure 5B*) show inter-individual variation in *P. rettgeri* proliferation. No individual *Dredd^EP142* mutant fly could control infection and all mutant hosts died with a high pathogen burden (*Figure 5B*). This demonstrates that the Imd pathway is required to control *P. rettgeri* infection and shift the infection to a persistent state in a subset of hosts.

The Toll pathway provides the primary response against Gram-positive bacteria but can also be activated by host damage and pathogen-derived virulence factors (*El Chamy et al., 2008*). We infected Toll-deficient (*spz^rm7* mutants) and wild-type hosts with *P. rettgeri* and observed the emergence of discrete high-load and low-load groups of flies of both genotypes during the first 16 hr of infection (*Figure 5C*). This demonstrates that Toll-deficient flies are sometimes able to control *P. rettgeri* infection. We next tested the role of the Toll pathway upon infection with a Gram-positive bacterium, *E. faecalis*, which also results in either acute lethality or chronic persistence in wild-type flies (*Figure 3C*). Without exception, flies deficient for the Toll pathway (*spz^rm7* mutants) failed to control *E. faecalis* infection and all hosts died with high bacterial loads (*Figure 5D*). These data collectively demonstrate that the specific and appropriate arm of the humoral immune response is critical for the control of bacterial infection and establishment of persistent infection.

Altogether, our data indicate that the capacity to control infection versus allowing it to proceed to the lethal state is determined by the action of the specific and appropriate arm of the immune response, but that the lethal pathogen burden is independent of immunity. This reinforces the idea that the *BLUD* is a readout of disease tolerance, but that the probability of the ultimate outcome of infection is mechanistically distinct and determined by host immunological control.

## A mixture model to capture within-host bacterial growth dynamics

We have empirically demonstrated that even genetically identical hosts differ in their ability to control infection, which ultimately leads to binary outcomes of death at high pathogen burden or survival with chronically persistent infection. In order to quantitatively define how the kinetics of bacterial proliferation and host response determine ultimate infection outcome, we developed a model to estimate bacterial growth within each individual host (illustrated in *Figure 6A*). Our model assumes that in the early phase of infection, the bacteria grow unchecked within the host at rate $\mu$. Empirical support for this component of the model comes from the observation that bacteria proliferate at the same rate as in in vitro culture over the first few hours of infection in a wild-type host and until death in an immune deficient host (as shown in *Figure 3—figure supplement 1*). We assume that there is a time post-injection where the immune response of an individual fly becomes sufficiently active to restrict bacterial proliferation, which we term the 'time to control' ($T^c$), and we define $\bar{t}^c$ as the average time to control of a given population. This is based on the empirical observation that an intact immune response is required for establishment of a persistent infection (see *Figure 5B*) and that the timing of our observed divergence in host trajectories corresponds roughly to the time at which the *D. melanogaster* humoral immune response begins to become active (*Lemaitre and Hoffmann, 2007*). We also observed that infections that kill only a fraction of the hosts reach a state where flies either have a high bacterial load, in which case they die, or a low bacterial load, in which case they survive. We model this divergence between the two types of bacterial kinetics by hypothesizing a threshold bacterial number ($n^{tip}$) that, if reached prior to control, results deterministically in ultimate host death at high bacterial load (the BLUD).

By applying our model to the empirical data, we can obtain for each individual host a probability corresponding to the likelihood that control is effective (see *Figures 3* and *5*, where white dots are infections that are most likely uncontrolled while red dots are most likely to be controlled). This probability is computed as the probability that bacterial load does reach $n^{tip}$ before control becomes effective. Using this computation, we estimated the overall probability that an individual host controls an infection by a given bacterium. We found that the estimated probability of control in our model, which is obtained from the measurements of bacterial load in different flies at different times along the infection (i.e. there was no repeated measurement on the host), correlates well with the empirically observed proportion of hosts that are killed (Spearman rank correlation test: S = 3.02, r = 0.97, p-value=8.17e-6; *Figure 6B*). This is both an indication that the model adequately describes our data and a confirmation that the probability of death is determined by the early phase of infection as described by the model.

One parameter that our model allows to estimate from experimental data on early growth is the average time to control ($\bar{t}^c$; *Figure 6C*). We hypothesized that 'priming' the immune system prior to infection should dramatically reduce this parameter and therefore yield much higher probability of controlling the infection. To test that, we infected transgenic flies that had constitutive activation of the Imd pathway with *P. rettgeri* and monitored bacterial growth. We found that bacterial loads in hosts with an immune system that was genetically activated prior to infection began to decrease

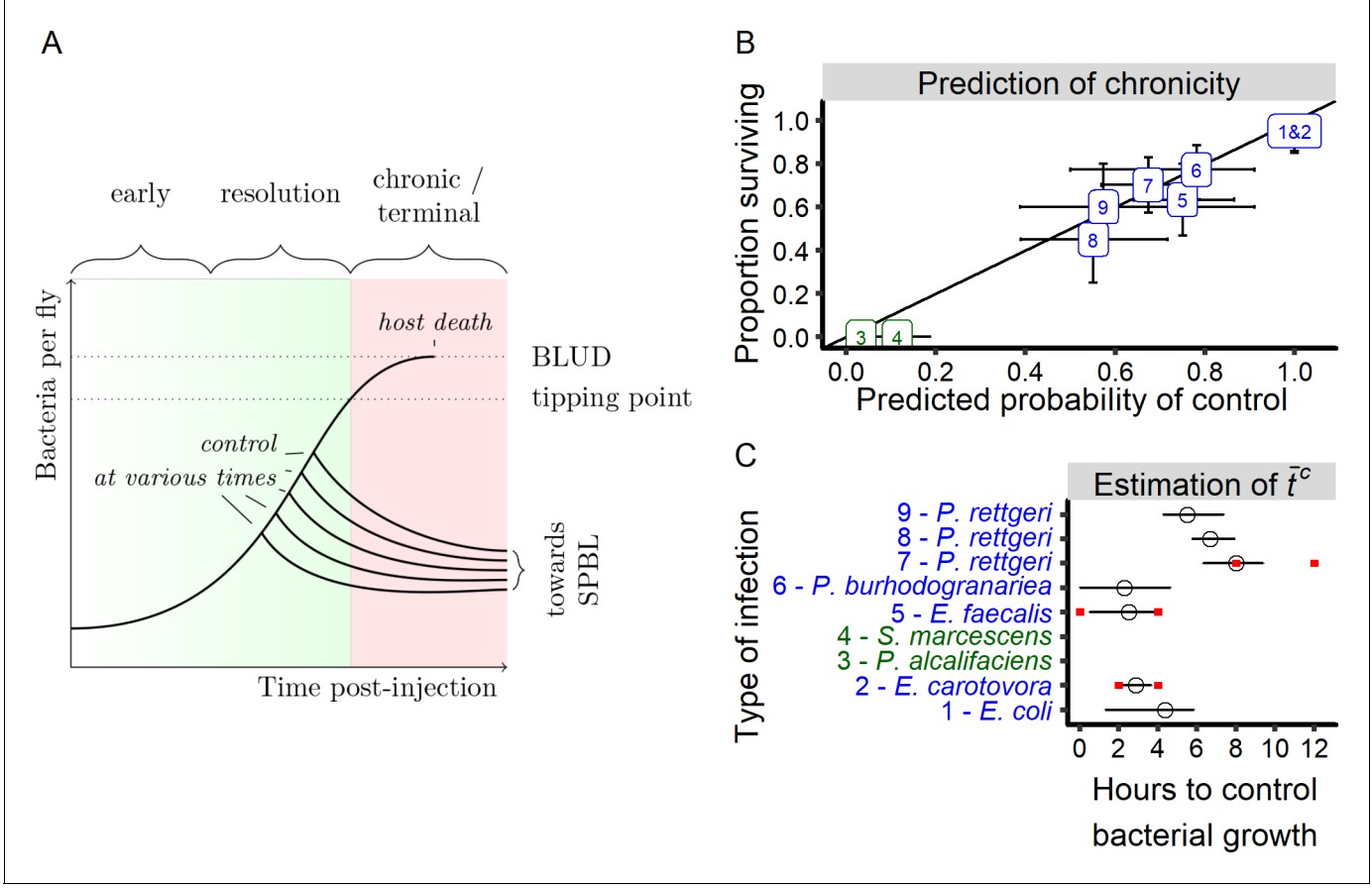

**Figure 6.** A generalized model of infection. (**A**) A schematic representation of a conceptual model for within-host bacterial growth dynamics that, upon three phases, lead either to host survival or death. In a first early phase, the bacterial population grow exponentially without being controlled. Then, in a second phase, some hosts start to control bacterial proliferation. In this resolution phase, inter-individual variation in the time to control (Tc), appears. We hypothesize that if host immune defenses control bacterial multiplication before bacterial load exceeds a critical threshold called the 'tipping point', then the host will survive infection and enters in a chronic phase, sustaining a persistent pathogen burden that we call the Set Point Bacterial Load (*SPBL*). If the tipping point is reached before the host establishes effective control, the infection enters a terminal phase and the bacteria will continue to proliferate until reaching a load that cannot be sustained by the host (the *BLUD*, or Bacterial Load Upon Death), at which point the host dies. (**B**) Prediction of the proportion of infected individuals that control their infections. The probability of controlling infection is estimated by our model based on early bacterial load measurements and correlates strongly with the proportion of hosts still alive when the mortality curve plateaued (Spearman rank correlation test: S = 3.02, r = 0.97, p-value=8.17e-6). The oblique line represents a 100% correlation. Error bars on the x-axis represent error of estimation by the model of the probability of controlling proliferation based on bootstrap analysis (1000 iterations). Error bars on the y-axis represent the confidence interval of the proportion of surviving hosts at the time where the mortality curve plateaued. The numbers and colors in the figure indicate the identity of the bacterium used in infection, and correspond to the numbers and identities in *Figure 3*. Green labels are bacteria that kill all the hosts and blue labels are bacteria that establish persistent infection in surviving hosts. (**C**) Estimation of $\bar{t}^c$ from early growth dynamic of different infections (*Figure 3*) with our statistical model. Red intervals are empirically estimated (see *Figures 5D* and *7B*). Black error bars on the x-axis represent error of estimation by the model of the $\bar{t}^c$ based on bootstrap analysis (1000 iterations).

DOI: https://doi.org/10.7554/eLife.28298.021

The following source data and figure supplement are available for figure 6:

**Source data 1.** Data set for *Figure 6*.
DOI: https://doi.org/10.7554/eLife.28298.023
**Figure supplement 1.** A simulation of the WHD model.
DOI: https://doi.org/10.7554/eLife.28298.022

immediately after injection, demonstrating effective control at the moment of infection ($\bar{t}^c$0) (*Figure 7A*). Surprisingly, however, the bacteria grew to acute lethality in a small proportion of individuals with constitutive Imd pathway activation (*Figure 7A*), suggesting that the Imd pathway alone is not always sufficient to control *P. rettgeri*.

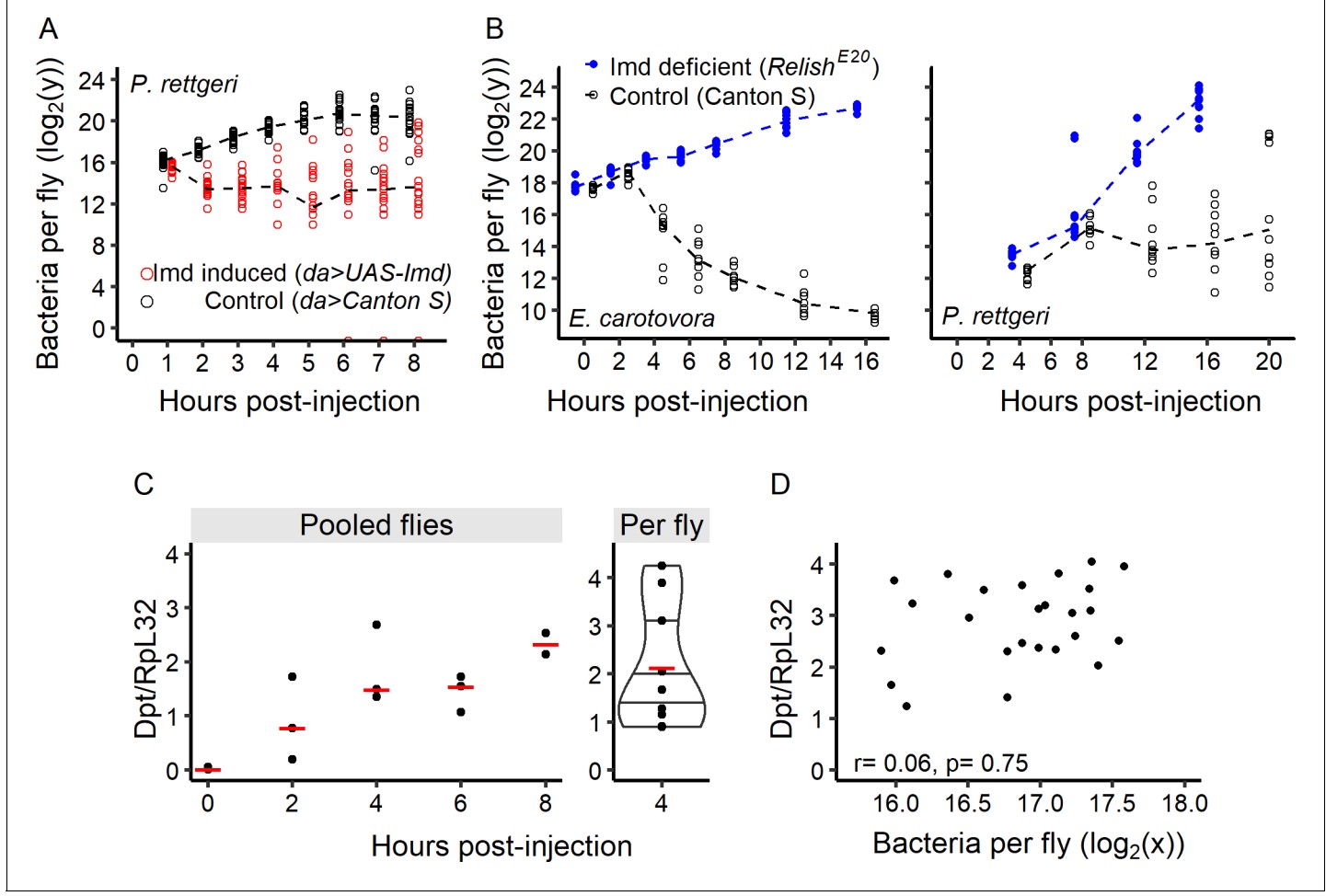

**Figure 7.** The time to effective control by the immune response determines the outcome of infection. (A) Bacterial load over time in hosts with pre-activated immune systems (*Gal80^{TS}; daughterless-Gal4 > UAS* Imd) and in the corresponding control flies (*Gal80^{TS}; daughterless-Gal4 > Canton* s). Early activation of immune response allows flies to start controlling bacterial growth immediately after the infection started. (B) Empirical determination of $t^e$. For both Gram-negative bacteria tested, within-host bacterial proliferation was initially similar between lines (GLM with Gamma distribution of error, *Time x Line*: df = 1, F = 0.4, p=0.53 for *P. rettgeri* up to 8 hr post-injection, and up to 2 hr for *E. carotovora Time x Line*: df = 1, F = 0.27, p=0.61), and later was controlled in wild-type (Canton S) but not in Imd-deficient flies (*Relish^{E20}* mutants). (C) The antimicrobial peptide gene *Diptericin* (measured relative to housekeeping gene *Rpl32*) in Canton S flies was strongly induced after injection of *P. rettgeri*. This can be seen in three replicate pools of flies tested every two hours post-injection (left panel with red lines representing median values), as well as for eight individual flies sampled 4 hr post-injection (right panel). (D) Correlation between bacterial load in individual fly heads (correlated with bacterial load in the abdomen, see *Figure 7—figure supplement 1*) and *Diptericin* expression in Canton S flies 8 hr post-injection with *P. rettgeri*. The lack of a significant correlation (Spearman correlation test, S = 4225.9, r = 0.06, p=0.75) suggests that the activation of the immune system is independent of early bacterial load. Dashed lines in panels A and B represent the connection between medians.

DOI: https://doi.org/10.7554/eLife.28298.024

The following source data and figure supplement are available for figure 7:

**Source data 1.** Data set for *Figure 7*.
DOI: https://doi.org/10.7554/eLife.28298.026

**Source data 2.** Data set for *Figure 7—figure supplement 1*.
DOI: https://doi.org/10.7554/eLife.28298.027

**Figure supplement 1.** Correlation between the number of *P. rettgeri* in head and abdomen of the same fly (Spearman correlation test, S = 633.31, r = 0.84, p=8.8e-09).
DOI: https://doi.org/10.7554/eLife.28298.025

We compared the average time to control ($t^c$) estimated from the model with experimental measurements of the time at which host immunity impacts bacterial load after infection with the same bacteria. We quantified proliferation of *E. carotovora* and *P. rettgeri* in wild-type and Imd pathway mutant hosts (*Figure 7B*) and the growth of *E. faecalis* in wild-type and Toll pathway mutant hosts (*Figure 5D*), using the logic that bacterial loads should become measurably different between wild-type and immune deficient flies soon after control by the immune response occurs in the wild-type host. Using this method, we found that the effect of the immune response is detectable starting at 2 hr post-injection for *E. carotovora*, at 8 hr post-injection for *P. rettgeri*, and at 2 hr post-injection for *E. faecalis* (*Figures 7B* and *5D*). These empirical measurements of $t^c$ ranges overlap with the confidence intervals of $t^c$ that our model predicts for these infections (*Figure 6C*).

We found variation in $t^c$ across bacterial infections, and we next hypothesized that this variation, because it impacts the probability of control ($P^c$), could partly determine the outcome of infection. In our model, the probability of control also depends on parameters that determine early bacterial growth. To analyze the contribution of $t^c$ and bacterial growth rate to the outcome of infection, we compared the likelihood (i.e. how well the model fits the data) of a complete model fitting the data through the estimation of 11 parameter values for each type of infections to the likelihood of models where some of the parameters are held constant for all infection types. For example, to estimate the role of the parameter $t^c$, we compared likelihood of our full model, where each type of infection has its own value of $t^c$, with a model fitting our data by estimating a common $t^c$ value for all infection types. A significantly better fit of the full model reveals a significant role of the parameter which was held constant in the second model. We performed this analysis for all the parameters involved in modeling the control of infection (Control model, i.e. $t^c$, $V^c$, $n^{tip}$) or the parameters involved in modeling early bacterial growth (Growth model, i.e. $t^{lag}$, $\mu$, $\sigma_b$) to test their relative impact on how accurately the model predicts survivorship (*Table 2*). We found that both sets of parameters involved in modeling the control of infection and early bacterial growth both have a strong impact on adequately predicting bacterial load (*Table 2*). We found that variation in survivorship is best explained by variation in $t^c$ when only bacteria displaying intermediate survival are considered (i.e. when excluding *E. coli*, *E. carotovora* which have very low growth and mortality rates, and *S. marcescens* and *P. alcalifaciens* which have high growth and mortality rates, see *Figure 6B*). Conversely, when all bacteria were included in the analysis, differences in survivorship are best explained by differences in early bacterial growth (*Table 2*). These results suggest that the probability of surviving infection is affected by the average time at which control occurs ($t^c$), but that this effect can be masked by large differences in growth rates ($\mu$).

**Table 2.** Effect of early growth ($t^{lag}$, $\mu$ and $\sigma_b$) and control ($t^c$, $V^c$ and $n^{tip}$) parameters on the capacity of the model to predict survival. Log-likelihood is computed either on bacterial load data (i.e. on the data set we used to fit the model) or on survival data. In this latter case, we used the probability of control predicted by the model as an offset in a binomial general linear model (glm) fitted to survival data. This glm includes an intercept which quantifies the difference between predicted control probability and observed survival. p-values indicate whether keeping a parameter constant for all bacteria significantly alters survival prediction.

| | | Complete data set | | | Intermediate survival | | |
| | | Bacterial load | Survival | | Bacterial load | Survival | |
| | df | logLik | logLik | p-value | logLik | logLik | p-value |
|---|---|---|---|---|---|---|---|
| Full | | −1243.119 | −17.85860 | | −800.0603 | −11.48770 | |
| Control | 24 | −1629.965 | −21.68408 | 0.999 | −839.5523 | −17.18319 | 0.496 |
| $t^c$ | 8 | −1660.396 | −19.18941 | 0.954 | **−839.5791** | **−16.82534** | **0.030** |
| $V^c$ | 8 | −1643.985 | −17.37663 | 1 | −826.4296 | −14.67760 | 0.173 |
| $n^{tip}$ | 8 | −1634.382 | −25.23397 | 0.064 | −832.8896 | −12.85596 | 0.603 |
| Growth | 24 | **−1798.630** | **−81.13355** | **6.512e-16** | −854.1100 | −11.37739 | 1 |
| $t^{lag}$ | 8 | −1692.743 | −16.49076 | 1 | −835.3833 | −11.30397 | 1 |
| $\mu$ | 8 | **−1705.175** | **−61.56015** | **1.564e-15** | −836.3794 | −11.75406 | 0.970 |
| $\sigma_b$ | 8 | −1704.156 | −21.95270 | 0.415 | −848.2682 | −12.28487 | 0.810 |

DOI: https://doi.org/10.7554/eLife.28298.028

## Variation in $T^c$ and inter-individual variation in infection outcome

Our analyses so far indicate that variation in infection outcome is strongly dependent on $t^c$, which should be determined by both the kinetics of the host immune response and the bacterial sensitivity to host immunity. For infection by a given bacterium, with a known and constant $t^c$, our model predicts that inter-individual variation in infection outcome must stem from variance in time to control ($T^c$), quantified by $V^c$. Setting this variance to zero for all infections indeed produces a probability of control which is either zero, with all individuals dying from infection, or one, with all infections being controlled. Since we see empirical variability in control of some infections, we hypothesized this might arise from biological variance in $T^c$.

We empirically tested the hypothesis that the speed and magnitude of Imd pathway induction might vary among individual flies, potentially resulting in variation in $T^c$ that leads to variation in ultimate infection outcome. To test this, we first determined the kinetics of Imd pathway activation after infection with *P. rettgeri* by quantifying mRNA levels of the *Diptericin* gene, which encodes an antimicrobial peptide and provides an excellent read-out of Imd pathway activity (*Buchon et al., 2014*; *Lemaitre et al., 1997*). We began by measuring *Diptericin* expression in pools of 20 flies Canton S. *Diptericin* expression was extremely low at the time of injection (*Figure 7C*, left), but a measurable induction was detected at 2 hr post-injection and its level continuously increased during the first 8 hr post-injection (up to ~100 fold induction compared to the start of the infection). Thus, the Imd pathway is activated transcriptionally very early upon infection, well before we detected any impact on bacterial proliferation. To test whether inducibility is variable among individual hosts, we quantified *Diptericin* expression level in single flies at 4 hr post-injection (*Figure 7C*, right). Time points equal or earlier than 8 hr were chosen because they occur before $t^c$ and before we begin to see inflated variance in the bacterial load of individual flies. At later times of infection, higher bacterial number leads to increased immune system activation in a positive feedback loop, so variation in AMP gene expression level becomes an indicator of the differences in bacterial loads instead of an estimation of the variance of induction in response to a unique bacterial load. At 4 hr post-injection, however, immune induction can be detected (*Figure 7C*) but immune control is not yet effective (*Figure 7B*) and no divergence in bacterial load is yet observed (e.g. *Figure 3D and E*). At 4 hr post-injection, we detected strong among-individual variation in *Diptericin* expression, ranging from little induction to strong immune response, with approximately 5-fold difference between extremes (*Figure 7C*), which may result from experimentally undetectable differences in handling, infection, or developmental history. We then did paired measurements of AMP expression and bacterial loads from single flies at 8 hr post inoculation, which is just before the predicted $t^c$. The Imd pathway is activated and *Diptericin* is expressed primarily in the *D. melanogaster* fat body, most of which is located in the abdomen, and we determined that the bacterial burden in the head of a fly is tightly correlated with the load estimated from the thorax and abdomen: ($r = 0.84$, p=8.8e-9; *Figure 7—figure supplement 1*), so we tested whether pathogen burden estimated from the heads of individual flies could be predicted by the *Diptericin* expression measured from abdominal RNA extractions. We found no significant correlation between bacterial load and the level of *Diptericin* transcripts 8 hr after injection (Spearman correlation: $r = 0.06$, S = 4225.88, p=0.75). This indicates that empirical variability in Imd activation is independent of variation in bacterial load at the early stage of infection. The disconnect may allow the infections of some hosts to grow through the critical threshold $n^{tip}$ prior to $t^c$, ultimately leading to host death, while the seemingly identical infections of other hosts are effectively controlled.

## Discussion

We evaluated infection of *Drosophila* by multiple bacteria that cover a range of pathogenicity, ranging from completely lethal infections that kill all hosts to fairly benign infections where most or all hosts survive. Most interestingly, we demonstrate binary outcome of infection with pathogens such as *P. rettgeri* or *E. faecalis*. Even among hosts that are identical in genotype, sex and age and are raised in a common environment, a fraction of hosts die with high bacterial burden while the remainder survive indefinitely with persistent low pathogen load. We identify a set of key parameters that determine which of these outcomes occurs and we propose a general model framework to describe the dynamic interplay between host and pathogen. We find that host death is associated with a bacterial load, the *BLUD*, which varies with host genotype and pathogen strain but does not vary with

inoculum dose. The *BLUD* does not correlate with the time a host will die from infection. Instead, we suggest that inter-individual variation in the probability of death stems from variation in the time it takes for each individual fly to establish effective immunological control of the infection ($T^c$), with the understanding that this must occur before the pathogen reaches a critical density ($n^{tip}$) if control is to be achieved. Hosts that survive their infections do so carrying a fixed bacterial load, the *SPBL*.

Based on the quantification of bacterial load in the host, we have built a model that describes inter-individual variability in bacterial growth (*Figure 6A*). In an early phase of the infection by pathogens, bacterial proliferation occurs at a rate similar to that of in vitro growth, suggesting that growth is essentially unconstrained in the first hours of infection. After a brief delay, the host immune response becomes sufficiently active so as to limit or perhaps even reverse bacterial growth. This time, which we parameterize as $t^c$ in our model, will certainly depend on both the kinetics of immune induction by the host and the sensitivity of the pathogen to the host immune response. In a following phase of our model, the infection resolves in one of two ways. The bacteria may continue to proliferate at the unconstrained rate until the host dies at a lethal load, the *BLUD*, or the bacteria may cease proliferation and settle into a phase of long-term persistent infection with a burden defined as the *SPBL*. In principle, the *SPBL* could be complete elimination of the bacteria, although we very rarely see that with *D. melanogaster* hosts. We infer that the probability of an infection entering into the lethal versus persistent state is defined by the probability that control is effective before bacteria grow above the critical threshold density, $n^{tip}$. Thus, the probability of lethal infection is most influenced and therefore predicted by the rate of bacterial proliferation ($\mu$) and the average time required to mount an effective immune response, $t^c$.

The simplicity of our statistical model yields some crucial insights and implications. First, a complex and apparently stochastic organism-level phenotype (survival versus death) can be satisfactorily predicted by modeling the probability of transitions between a few discrete states of infection (early growth, uncontrollable growth, and decrease/stabilization after control). In the course of infection, each may occur in discrete temporal phases (termed 'early phase', 'resolution phase', 'terminal phase' or 'chronic phase' in *Figure 6A*) that have distinct and possibly deterministic outcomes, as those described in (*Duneau et al., 2011*; *Ebert et al., 2016*; *Hall et al., 2017*; *Levin and Antia, 2001*; *Schmid-Hempel and Ebert, 2003*). Furthermore, each of these states and the parameters that determine them has a distinct mechanistic underpinning and may be differentially subjected to natural selection. For example, host evolution of shortened time to control through increased inducibility of the immune response would have a different genetic basis than one that involved reduced bacterial proliferation through host sequestration of nutrients (e.g. metal ions or carbon). Both mechanisms might result in apparent 'resistance' to infection, both would contribute to increased probability of survival from infection, and each would carry distinct costs or tradeoffs. From the pathogen's perspective, changes in its growth rate or sensitivity to host immune responses would also alter the probability that they become lethal versus persistent, also with distinct mechanistic bases and costs or tradeoffs. Each predictive parameter in the model can be experimentally manipulated, and the entire model can be theoretically evaluated to identify evolutionary optima in different environments and under different transmission requirements. We note that this model can be easily extended to infection by other classes of pathogen, to specific tissues such as the gut, or to hosts that display secondary immune responses (e.g., antibody-mediated acquired immune responses).

Interestingly, the two endpoint parameters of infection, bacterial load in survivors (*SPBL*) and in hosts that succumb (*BLUD*), were remarkably invariant among individual hosts of a same genotype even though they varied substantially across infections by different pathogens, host species and genotypes. The mechanisms underlying the death of a fly upon infection have been curiously overlooked, and the nature of damage to *D. melanogaster* from bacterial infection remains elusive (*Dionne and Schneider, 2008*). We find that flies die at a specific bacterial load that does not depend on initial infection dose or eventual time to death. This implies that death is unlikely to be a direct consequence of accumulated infectious damage, since longer term exposure to low pathogen burdens might trigger as much damage as short-term exposure to high bacterial burden. Instead, our results are more compatible with a model for death occurring as a consequence of sepsis or multiple organ failure driven by pathogen load (*Baue, 1975*). Of course, the lethal load of different pathogens would vary depending on the toxicity of the particular pathogen. In that sense, we can argue that the *BLUD* is a measure of pathogenicity of the bacterium, as it represents the minimal

density of bacteria that kills a host. We hesitate to use the word 'virulence' though, because the *BLUD* does not correlate with the speed at which pathogens kill or with the probability of host death. Reciprocally, the *BLUD* also represents a measure of host disease tolerance (*Soares et al., 2017*) as it corresponds to the maximal bacterial load that can be sustained in the host without dying from the infection.

Our results are somewhat inconsistent with those of *Haine et al., 2008*, which documented rapid cellular elimination of bacteria injected into *Tenebrio molitor* and posited an only minor role for the humoral response. In contrast, our study indicates that the timing and intensity of AMP production is the most crucial factor for controlling bacterial infection in *Drosophila*. One intriguing hypothesis for reconciling the two observations is that adult *Drosophila* have only a small number of hemocytes due to loss of hemocytes upon aging (*Guillou et al., 2016*), while this might not be the case in *Tenebrio*. Thus, the relative importance of cellular versus humoral immunity may differ between the two insects. Additionally, different bacterial pathogens were employed in the two studies, and different microbes may be differentially sensitive to alternative arms of the immune response so generalizations should be made with caution.

We found that complete bacterial clearance rarely happens in *Drosophila*. With all seven pathogens in this study, in no case were the bacteria fully cleared from all hosts even a week after the initial infection, even in cases where 100% of hosts survived the infection. Persistent infections stabilized at a fixed load for each pathogen, the *SPBL*. On rare occasion, individual flies in our study would carry persistent infections for several days before suddenly dying with a pathogen burden that had reached the *BLUD*. This suggests that there are some conditions under which a persistent infection can re-emerge as an acute infection. This is reminiscent of chronic infections by multiple human pathogens, including HIV, eukaryotic parasites such as *Toxoplasma gondii*, and bacteria such as *Salmonella enterica*, *Mycobacterium tuberculosis* or *Streptococcus pneumoniae*. These parasites are considered to be 'specialists' at chronic infection, yet our results suggest that qualitatively similar phenomena can be obtained from infection with a broad range of generalist bacteria. As the *SPBL* varies with both host and pathogen genotype, it is presumably subject to selection in both. In that context, it is worth noting that forcing an infection into the persistent state is effectively a disease tolerance strategy from the host perspective, yet the pathogen burden borne may still carry lifelong fitness cost. From the pathogen's perspective, either persistence or acute lethality may be favored depending on the conditions of transmission. Moreover, selection for high growth rate due to competition between bacterial genotypes within a host (*Alizon et al., 2009*; *2013*) may trade off against the capacity to establish persistent infection.

*Levin and Antia (2001)* suggested that 'us and other living organisms are little more than soft, thin-walled flasks of culture media'. With the present work, we argue that hosts are slightly more complex and that potentially minute variations in host or pathogen physiology during the early phase of infection can have dramatic effects on the ultimate outcome. We have defined three parameters ($t^c$, $\mu$, $n^{Tip}$) that are sufficient to predict ultimate infection outcome, although we still do not know whether variation in those parameters is due to micro-environmental variation, uncontrolled plasticity in developmental history, somatic mutations, or other factors that are uncontrolled or uncontrollable in experiments (e.g. depth of penetration of the needle during injection, site at which bacteria accumulate etc.) and in nature (e.g. time since last meal or mating, psychological status of the fly etc.). Nevertheless, what is becoming clear is that small differences in pathogen infectivity and host immunological control, especially in the early stages of infection, may manifest as large differences in the outcome of infection.

## Materials and methods

### Fly stocks and husbandry

*Drosophila melanogaster* were reared on glucose-yeast medium (82 g/L yeast, 82 g/L glucose, 1% *Drosophila* agar, supplemented with 2.5 mg/L methylparaben and 10 mL of a solution of phosphoric and propionic acid: 41.5 ml Phosphoric Acid + 418 mL Propionic Acid + 540.5 mL distilled water). At day two after eclosion, adults were isolated in groups of five males and five females. All experiments were conducted with mated males 5 to 8 day post-eclosion. Rearing and experiments were conducted at 25°C (±1°C) with a 12 hr light/dark cycle. Canton S, Oregon R and w[1118] were used as

wild-type laboratory strains. Eight fully isogenic lines from the *Drosophila* Genome Reference Panel (DGRP: RAL-85, RAL-138, RAL-359, RAL-882, RAL-594, RAL-707, RAL-712, RAL-714) were randomly chosen and used to assay the role of inter-individual genetic variation on several infection parameters (*Mackay et al., 2012*). All wild-type stocks and isogenic lines were made axenic then re-associated with a controlled mixture of 5 bacterial species that compose the core microbiota of *Drosophila* (*Wong et al., 2011*) to ensure homogeneity of associated microbiological communities. Briefly, eggs of these stocks were sterilized by bleaching and reassociated with a mixture of 5 common gut microbes of *Drosophila* (*Acetobacter pomorum*, *Acetobacter tropicalis*, *Lactobacillus plantarum*, *Lactobacillus brevis*, *Lactobacillus fructivorans*) as previously published. Immune mutant stocks have been described previously and include mutants of the Imd pathway (*PGRP-LE*$^{112}$, *PGRP-LC*$^{\Delta E}$ double mutants (*Takehana et al., 2004*), *Dredd*$^{EP1412}$ (RRID:BDSC_10456, *Leulier et al., 2000*) and *Relish*$^{E20}$ mutants (RRID:BDSC_55714, *Hedengren et al., 1999*), mutants of the Toll pathway (*spz*$^{rm7}$, [*Lemaitre et al., 1996*]), and mutants of the melanization cascade (double mutant *PPO1Δ*, *PPO2Δ*, [*Binggeli et al., 2014*]). We used the *Gal80*$^{TS}$; *daughterless-Gal4* driver in combination with *UAS-Imd* to ubiquitously induce the Imd pathway at the adult stage. To generate phago$^{less}$ adult flies, we ablated phagocytes by inducing the pro-apoptotic gene *Bax* in hemocytes using the hemocyte-specific driver *Hml-Gal4* (*Hml-Gal4 >UAS* GFP [RRID:BDSC_30140]; *UAS-Bax, Gal80*$^{TS}$) in adult flies at 2 days post-eclosion (*Defaye et al., 2009*).

## Bacterial infection

The bacterial strains used in this study include the Gram-negative *Providencia rettgeri* (strain Dmel) (*Juneja and Lazzaro, 2009*), *P. burhodogranariea* (*Juneja and Lazzaro, 2009*), *Serratia marcescens* (strain Db11) (*Kurz et al., 2003*), *E. coli* (Type strain), *Pectinobacterium carotovorum carotovorum 15* (strain *Ecc15-GFP*; formerly genus *Erwinia*, [*Basset et al., 2000*]), *Pseudomonas entomophila* (*Vodovar et al., 2005*), and *Providencia alcalifaciens* (*Juneja and Lazzaro, 2009*), as well as the Gram-positive *Enterococcus faecalis* (isolated from wild-caught *D. melanogaster* by B. Lazzaro), and *Staphylococcus aureus* (strain PIG1 [*Liu et al., 2005*]). Cultures were grown to saturation overnight at 37°C (29°C for *E. carotovora*) in LB liquid medium (LB broth, Miller, VWR). Saturated cultures were suspended and diluted in Phosphate Buffered Saline (PBS, pH 7.4) to the desired optical density (OD$_{600}$ = 1 or as otherwise indicated). We injected 23 nL of bacterial suspension (c.a. 30,000 bacteria for OD$_{600}$ = 1) into each fly abdomen using a Nanoject II (Drummond) (*Khalil et al., 2015*). Flies were anesthetized with light CO$_2$ for about five minutes during the injection procedure and were observed shortly after injection to confirm recovery from manipulations. We measured host survivorship post-injection in groups of 20 or 50 males kept with *ad libitum* access to food. Differences in survivorship were tested with Cox regression models (R package Survival).

## Within-host bacterial load dynamics and *BLUD* estimation

To characterize the dynamics of within-host bacterial loads, after being dipped in Ethanol 70% and washed in PBS to limit contamination by external bacteria, individual flies were homogenized in 500 μl of sterile PBS with an HT homogenizer (OPS Diagnostics) at each timepoint post-injection. The homogenate was then diluted to 1:100 or 1:1000 in PBS to ensure that plate counts remained within the limits of resolution of the plating system. We plated 70 μl onto LB agar using a WASP II Autoplate spiral plater (Microbiology International). Plates were incubated overnight at 37°C (29°C for *E. carotovora*) and bacterial colonies were counted using an EZ-Count Automated Colony Counter (Microbiology International) to estimate the number of viable bacteria per fly. To estimate the Bacterial Load Upon Death (*BLUD*), infected hosts were checked every 30 min and newly dead flies (flies not moving and on their side or back) were collected and homogenized, with bacterial load quantified as described above.

## Gene expression by RT-qPCR

Total RNA was extracted from either 20 flies or single flies with TRIzol reagent (Invitrogen). Template RNA (1 μg) was used to generate cDNA by reverse transcription using the SuperScript II cDNA synthesis kit (Promega) and then analyzed by quantitative polymerase chain reaction (qPCR) using the PerfeCTa SYBR Green SuperMix (Quantabio). Expression values were normalized to *RpL32*. Primer sequences are available in method supplementary 1. At least three independent repeats were done.

## Statistical analyses

We used non-parametric tests to compare bacterial loads between groups of flies: Wilcoxon for two groups comparison and Kruskal-Wallis for more than two groups. The only exception was the comparison of loads between immune deficient mutants and wild-type controls, where, to take into account that we used more than one genotype to test the same biological hypothesis, we used a linear regression with genotype as a random effect. We used Spearman correlations to assess the relationship between bacterial loads and several quantitative traits (initial inoculum, time to death, and *Dpt* expression). All data analysis was performed using R version 3.3.1 (*R Core Team , 2017*).

## Model of growth dynamics

Our results demonstrate that experimentally identical hosts exposed to the same pathogenic infections can either control the infection or succumb to acute bacterial proliferation. In addition, we see that bacterial growth in the absence of immune control has a similar rate to that of in vitro LB cultures (see Results). We therefore built a model that consists of a mixture of two demographic models, one describing within-host bacterial growth and one modeling bacterial elimination (see *Figure 6A* for a representation of the conceptual model this statistical model is based upon). More precisely, we considered that each bacterial load estimated on a single fly ($n$) was a log-normal random variable with the mean computed from a Baranyi model (*Baranyi and Roberts, 1994*) when bacteria grow in the host uncontrolled by the fly's immune system with

$$n_t = n^{max} \frac{e^{\mu t} - 1 + e^{\mu t^{lag}}}{e^{\mu t} - 1 + e^{\mu t^{lag}} \frac{n^{max}}{n_0}}$$

or from an exponential decrease model, when bacteria are controlled by the host, with

$$n_t = 1 + (n^c - 1)e^{-\delta t}$$

With increasing time, $t$, $n$ begins at $n_0$ and approaches $n^{max}$ in the Baranyi model, while it decreases from $n^c$ to asymptotically reach zero in the exponential decrease model. Actual bacterial load was assumed to follow a log-normal distribution, with average given by either of the two previous equations and variance fixed at $\sigma^2_b$ for the Baranyi model and $\sigma^2_d$ for the exponential decrease.

To reflect the probabilistic nature of the infection process, we developed a model to compute the probability that bacteria are either controlled (and thus have an exponential decrease) or are not (and thus have a Baranyi growth dynamic). We considered that the time required for the host immune defense to efficiently control bacterial infection follows a Gamma distribution with a fixed average $t^{lag} + t^c$ and a variance $V^c$. $t^c$ is the average time host defenses take to control bacteria after they have started to grow ($T^c$), and $V^c$ quantifies inter-individual variation in $T^c$. We further assumed that the host cannot control infections after the bacterial load reaches a fixed threshold, which we termed the tipping point ($n^{tip}$). The eleven parameters of this model are all described in *Table 1*.

Under the model, the probability that a host survives infection is the probability that the control of bacterial growth occurs before the tipping point is reached. More precisely, if $P^c$ is the probability of survival, $t^{tip}$ is the time at which the tipping point is reached, and $T^c$ is a random variable corresponding to the time of control observed for a given host, we can write

$$P^c = Pr(T^c < t^{tip})$$

The probability of survival $P^c$ therefore depends on parameters that determine how fast bacteria grow initially ($n_0$, $n_{max}$, $t^{lag}$, $\mu$), on the tipping point itself ($n^{tip}$), and on parameters, $t^c$ and $V^c$, that govern the distribution of time to control $T^c$.

All these parameters can be estimated by adjusting the model on experimental data consisting of bacterial loads measured over time in different individuals. This was performed by computing the likelihood of the model for each bacterial load $x$ observed at time $t$ as

$$L(log_2 x, t) = L_b(log_2 x, t)(1 - p_t) + L_d(log_2 x, t)p_t$$

where $L_b(log_2 x, t)$ is the likelihood under the Baranyi model and $L_d(log_2 x, t)$ the likelihood under the exponential decrease model. The global log-likelihood of the model is then obtained by summing log-likelihoods computed for each point of the datasets. This was possible because each point

**Table 1.** List of parameters of the mixture model with their signification.

**Baranyi model**

| | |
|---|---|
| $n_0$ | Bacterial load upon injection |
| $n^{max}$ | Maximum bacterial load |
| $t^{lag}$ | Lag time |
| $\mu$ | Early bacterial growth rate |
| $\sigma_b$ | Standard deviation of loads in the absence of control |
| **Exponential model** | |
| $n^c$ | Intercept of the exponential decrease model |
| $\delta$ | Decrease rate in bacterial load when infection is controlled |
| $\sigma_c$ | Standard deviation of loads in controlled infections |
| **Control** | |
| $t^c$ | Average time to control |
| $V^c$ | Variance in time to control |
| $n^{tip}$ | Bacterial load above which the host cannot control infection |

DOI: https://doi.org/10.7554/eLife.28298.029

of our data set consists of a single individual host, and all measurements at all sampling times are therefore somewhat independent from each other (i.e. the measurements are not repeated on the same fly). We illustrated simulated outcomes of this model of within-host dynamics in *Figure 6—figure supplement 1*.

We estimated the 11 parameters of this mixture model by maximizing log-likelihood, using the Optim procedure in R. Confidence intervals for parameter estimates were then obtained by bootstrapping data for each experimental time point. This model, once adjusted, allows estimation of the probability that a host controls an infection by computing the probability that effective control occurs before $n^{tip}$ is reached. Probability of control therefore depends on the parameters of the Baranyi model ($n_0$, $n_{max}$, $t^{lag}$, $\mu$) and of those describing the distribution of time to control ($t^c$, $V^c$, $n^{tip}$).

# Acknowledgements

We thank Patrícia Beldade, Gary Blissard, Jeffrey Hogdson, Marine Cambon, Kathleen Gordon, Vanika Gupta, and Joo Hyun Im for comments on the manuscript, Hidetoshi Inamine for thoughtful discussions. We thank Bruno Lemaitre and Luis Teixeira for fly stocks. This work was funded by NSF 1354421, NSF 1656118 and funding from the Empire State Stem Cell Fund through New York State Department of Health NYSTEM contract C029542 for NB, NIH grant R01 AI083932 to BPL and a fellowship from the Swiss National Foundation (P300P3_147874) and from the French Laboratory of Excellence project 'TULIP' (ANR-10-LABX-41; ANR-11-IDEX-0002–02) to DD.

# Additional information

## Funding

| Funder | Grant reference number | Author |
|---|---|---|
| National Science Foundation | 1354421 | Nicolas Buchon |
| National Institutes of Health | RO1 AI083932 | Brian P Lazzaro |
| New York State Department of Health | Empire state stem cell fund C029542 | Nicolas Buchon |
| Schweizerischer Nationalfonds zur Förderung der Wissenschaftlichen Forschung | Fellowship from P300P3_147874 | David Duneau |

| Agence Nationale de la Recherche | French Laboratory of Excellence ANR-11-IDEX-0002-02 | David Duneau |
|---|---|---|
| Agence Nationale de la Recherche | French Laboratory of Excellence project ANR-10-LABX-41 | David Duneau |
| National Science Foundation | 1656118 | Nicolas Buchon |

The funders had no role in study design, data collection and interpretation, or the decision to submit the work for publication.

### Author contributions
David Duneau, Conceptualization, Data curation, Formal analysis, Funding acquisition, Methodology, Writing—original draft; Jean-Baptiste Ferdy, Conceptualization, Formal analysis, Methodology, Writing—original draft; Jonathan Revah, Hannah Kondolf, Gerardo A Ortiz, Data curation; Brian P Lazzaro, Conceptualization, Funding acquisition, Methodology, Writing—original draft; Nicolas Buchon, Conceptualization, Data curation, Funding acquisition, Methodology, Writing—original draft

### Author ORCIDs
David Duneau http://orcid.org/0000-0002-8323-1511

### Decision letter and Author response
Decision letter https://doi.org/10.7554/eLife.28298.032
Author response https://doi.org/10.7554/eLife.28298.033

## Additional files

### Supplementary files
• Supplementary file 1. Primer sequences used in the qPCR analysis.
DOI: https://doi.org/10.7554/eLife.28298.030
• Transparent reporting form
DOI: https://doi.org/10.7554/eLife.28298.031

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
