## [Decision Letter]

Thank you for submitting your article "Stochastic variation in the initial phase of bacterial infection predicts the probability of survival in *D. melanogaster*" for consideration by *eLife*. Your article has been favorably evaluated by Wendy Garrett (Senior Editor) and three reviewers, one of whom, Bruno Lemaître (Reviewer #1), is a member of our Board of Reviewing Editors. The following individual involved in review of your submission has agreed to reveal their identity: Andrea Graham (Reviewer #2).

The reviewers have discussed the reviews with one another and the Reviewing Editor has drafted this decision to help you prepare a revised submission.

Summary:

This elegant and thorough study provides rigorous quantitative dissection, under controlled environmental conditions, of the contributions of host and pathogen genetics plus random individual variation – in this case, stochasticity in the infection process itself – to the dynamics and outcome of infection. The authors make excellent use of the tools available for *Drosophila*, including isogenic lines and immune knockouts, as well as a wide array of bacterial species. They are therefore able to quantify variation among pathogen species in odds of killing the host, in the set point bacterial load, and in the consistent bacterial burden at which death occurs (regardless of the time at which that threshold burden is reached, interestingly). Likewise, their extensive empirical data show variation among host species and isogenic lines in bacterial load upon death, which appear independent of inoculum size and immune deficiency. The authors go on to show that stochastic variation within many of their host-pathogen pairings leads to bimodal outcomes: death or chronic infection. The authors explain this bifurcation as a race between rates of bacterial proliferation and immune response induction. The evolutionary implications of this are clear – e.g., selection on rates of immune response induction or bacterial proliferation. Indeed, evolutionary biologists, ecologists, microbiologists, physiologists, and immunologists working in all sorts of biological systems will be intrigued and inspired by this array of findings.

There are, however, a number of important issues to clarify before considering this paper for publication.

Essential revisions:

1) Impact of initial conditions:

A) In particular, the overall conclusion is that variations in the timing of initiating the humoral immune response (*t^c^*) determine the ultimate fate of infected animal – either death or survival with chronic infection. One trivial possibility that is not directly addressed is that (small) differences in initial inoculation (with little or no variation in the *t^c^*) are actually responsible. This would be consistent with data present in Figure 2—figure supplement 1, and the mathematical model presented, without postulating variation in *t^c^*. It seems that empirically determined initial dose is only reported in one figure panel (Figure 5). Given the variation in the pulled capillary needle and the injection device, how can the authors be certain small variations don't tip the balance?

1B) While it looks like that all individual flies were treated identically, another parameter that is clearly different for each individual fly is the exposure to CO_2_ during manipulations/injection. It is probable that those flies that were injected first (had lower exposure to CO_2_) in the end were survivors, while those injected last (had longer exposure to CO_2_) succumbed. This should be experimentally addressed, considering that CO_2_ is known to increase *Drosophila* sensitivity to infections (Helenius et al., 2009).

Collectively, the authors should provide more information to support their statements that this is not small variation in the initial condition that explain the stochasticity. While the influence of CO_2_ might be easily tested, the impact of differences in the initial dose could be better characterized by adding more measurement of the initial dose for several bacteria. Another way could be to monitor stochasticity in infected flies with two related doses in the same setting (Dose 1 being always slightly superior of Dose 2) and see the existence of flies that die faster with Dose 2 compared to Dose 1).

2) Analysis of Immune response and bacterial growth:

The analysis of Imd signaling (Dpt induction) on single flies is a key part of this paper, yet is massively under-developed. Only one time point is shown, and given that transcript is measured, when mature AMP is the functional output, it is not at all clear that this one time point is relevant. Then, in the next panel, a different time point is examined to argue that Dipt levels do not correlate with bacterial loads. It can be argued that an earlier time point, such as 2 hours, is really the most relevant, as the model argues that the timing of the initiation of the response is deterministic, and perhaps at the initiating time point a correlation between load and response would be observed. Did the authors try to analyze immune response and bacterial growth in live flies using adequate fluorescent reporter (Dpt-mcherry, *P. rettgeri* GFP) ? While being less quantitative, this could bring another set of data that support their main conclusions.

On this line, the notion that the methods used here demonstrate bacterial growth in an individual fly (see language in Figure 3 legend) is grossly overstated. Clearly, as a population, the evidence clearly demonstrates bacterial growth in vivo. However, the bacterial growth within any individual fly cannot be assessed by these methods.

3) Clarity of the text and relevant information:

Many parameters affect infection such a temperature, sex, injection methods…. This information should be in the Results section not hidden in the Materials and methods. It is not clear what each survival graph shows, is it a representative result? How many times was it repeated? How many flies? Males or females? For bacterial load graphs, were all individual flies taken from one experiment (one biological replicate) or is it pooled from several replicates? It is often difficult to know how the experiments were done. All these details should be provided. Moreover, we could not exclude batch effect. This is actually suggested by Figure 4 with the new bacteria appearing more virulent.

4) All the data should be made publicly available to allow other scientists to use them. I would suggest to include all the raw data in excel sheets (bacterial count, survival data; biological repeats…).

5) The idea that the bacteria proliferate at the same rate in LB medium and flies should be documented for other strains.

6) Description of the model: more information on the model should be included to the main text. As it is written it is hard to understand this section of the paper. Moreover, even as written it is not fully and clearly explained to a non-expert in the field of modeling. For example, the meaning and import of mini graph of "Frequency vs. *t^c^* was unclear to one reviewer, and the relationship of *P^c^* to the area below "the" curve was likewise unclear, as there are 2 curves presented. Some parameters are introduced but described later: *V^c^* is shown in the figure but mentioned only in the Discussion. The confusion on the description of the model stem partly because of the high density at which series of hypotheses, tests and results are presented, and different sets of empirical results invoked (e.g., the third paragraph of the subsection “A mixture model to capture within-host bacterial growth dynamics”, which covers huge terrain very rapidly). Perhaps greater use of paragraph breaks or subheadings would help the reader navigate.

Globally, a longer paragraph with the model, the definition of each parameter could improve the impact of this paper.

[Editors' note: further revisions were requested prior to acceptance, as described below.]

Thank you for resubmitting your work entitled "Stochastic variation in the initial phase of bacterial infection predicts the probability of survival in *D. melanogaster*" for further consideration at *eLife*. Your revised article has been favorably evaluated by Wendy Garrett (Senior Editor), a Reviewing Editor, and one reviewer.

The manuscript has been improved but there are some remaining issues that need to be addressed before acceptance, as outlined below:

Both the external reviewer and the reviewing editor found that the authors have adequately answered to most of reviewers’ comments. As you can see from the reviewer 3, there is regret that the paper does not include in vivo monitoring of antimicrobial peptide gene expression to back up the model. Nevertheless, it is true that GFP reporters are slow and might not be the appropriate tool to monitor the immune dynamic at early time point. To address this in particular, we would recommend that you adjust your text, in both the Results and the Discussion, to make it explicitly clear that you are inferring a variance in the initiation of response, but do not (yet) have tools available to demonstrate it.

Reviewer #3:

This remains one of the most thought provoking papers I have read in recent years. I think it will be a landmark in the field.

That being said, I continue to struggle with the big issue, which is not directly addressed by the revisions. In short, the authors have inferred that the differences in the timing or strength of the initial immune response following infection determine the outcome (death at *BLUD* or chronic infection at *SPBL*). Surprisingly, they write in the rebuttal that they have been unable to measure AMP induction via fluorescent protein reporters at these early time points, at least in intact flies. Ideally, I would love some type of data that can probe this inference.

Another more significant issue, is (still) the presentation of the mathematical model. Lots is made of the probability calculation in Figure 6 and beyond, but the way this probability is determine is not explained in the Results section. It is presented in detail in the Materials and methods, but frankly the math is over my head. I would prefer some discussion of this computation directly in the Results section so as to make sense of Figure 6, for example.

Finally, the rationale for the time points used in 7B (4 hour) and 7C (8 hours) is presented, and argued compellingly, in the rebuttal but this logic is still missing from the actual text of the article. I would recommend including.

---

## [Author Response]

Essential revisions:1) Impact of initial conditions:A) In particular, the overall conclusion is that variations in the timing of initiating the humoral immune response (t^c^) determine the ultimate fate of infected animal – either death or survival with chronic infection. One trivial possibility that is not directly addressed is that (small) differences in initial inoculation (with little or no variation in the t^c^) are actually responsible. This would be consistent with data present in Figure 2—figure supplement 1, and the mathematical model presented, without postulating variation in t^c^. It seems that empirically determined initial dose is only reported in one figure panel (Figure 5). Given the variation in the pulled capillary needle and the injection device, how can the authors be certain small variations don't tip the balance?

We agree with the reviewers that a key premise for our work is that the initial conditions are as homogenous as possible. Our method of infection by injection allows precise, reproducible introduction of a quantified bacterial suspension into individual flies. The empirically determined initial dose (time zero) is reported in both panels of Figure 3, in 3B, 5B, both panels of 5C, 5D and in 7A. In all cases, the range of inoculation size is never larger than two log_2_ (i.e. two bacterial divisions) and it is generally within one log_2_ of difference. In experiments where we did not quantify bacterial load at time = 0 hours, we have a quantification 1hr post-injection (i.e. about one division after the injection) and we detect inter-individual variation in bacterial load lower than a log_2_ in these conditions too. From the experiments where we intentionally varied the dose (see Figure 2—figure supplement 1), we know that a 2-fold difference in initial dose is not sufficient to guarantee the outcome of infection. We performed experiments using a 100-fold range of infection doses (from optical density 0.01 to optical density 1 and saw the binary outcome of infection across this entire range. This indicates that variation in the number of bacteria injected cannot explain by itself the binary outcome of infection.

We cannot exclude the possibility that there is variability in the damage inflicted to the fly during the injection process, and it is possible that this contributes to the dual outcome of infection. We limited this possibility by injecting flies in the abdomen and using a very thin needle. This infection method has been shown to be less lethal than classical pinprick method (see Chambers et al. 2014 Infection and Immunity).

We believe our data demonstrate that differences in *t^c^* are central to the variation in infection outcome, and any variation in *t^c^* that arises because of minor differences in experimental handling is sufficiently small that we are still able to clearly detect differences between host genotypes and infections with different bacterial species. If the primary cause of variation in the outcomes was random experimental differences, we would expect no consistent differences among host genotypes or between bacteria with similar but non-identical killing kinetics.

1B) While it looks like that all individual flies were treated identically, another parameter that is clearly different for each individual fly is the exposure to CO_2_ during manipulations/injection. It is probable that those flies that were injected first (had lower exposure to CO_2_) in the end were survivors, while those injected last (had longer exposure to CO_2_) succumbed. This should be experimentally addressed, considering that CO_2_ is known to increase Drosophila sensitivity to infections (Helenius et al., 2009).

It is an intriguing possibility that differences in exposure to CO_2_ could generate differences in resistance to infection and thus explain the variability we observe, but we do not believe there is any cause for concern. In our design, each round of injection lasts a maximum 15 min on a CO_2_ pad, and all flies remain on the pad for the entire time the treatment group is being infected. Thus, the flies that are injected first do not have shorter exposure to CO_2_. Nevertheless, we quantified the effect of 15 min of CO_2_ exposure before or after injection, compared to 2min of CO_2_ exposure, thus covering the full experimental variation in CO_2_ exposure in our settings. A difference of 15min of exposure to CO_2_ did not affect the chance of survival, we therefore believe that CO_2_ exposure is not responsible for our phenotype. These results are now presented in Figure 1—figure supplement 1.

Collectively, the authors should provide more information to support their statements that this is not small variation in the initial condition that explain the stochasticity. While the influence of CO_2_ might be easily tested, the impact of differences in the initial dose could be better characterized by adding more measurement of the initial dose for several bacteria. Another way could be to monitor stochasticity in infected flies with two related doses in the same setting (Dose 1 being always slightly superior of Dose 2) and see the existence of flies that die faster with Dose 2 compared to Dose 1).

We replied to this concern above.

2) Analysis of Immune response and bacterial growth:The analysis of Imd signaling (Dpt induction) on single flies is a key part of this paper, yet is massively under-developed. Only one time point is shown, and given that transcript is measured, when mature AMP is the functional output, it is not at all clear that this one time point is relevant. Then, in the next panel, a different time point is examined to argue that Dipt levels do not correlate with bacterial loads. It can be argued that an earlier time point, such as 2 hours, is really the most relevant, as the model argues that the timing of the initiation of the response is deterministic, and perhaps at the initiating time point a correlation between load and response would be observed. Did the authors try to analyze immune response and bacterial growth in live flies using adequate fluorescent reporter (Dpt-mcherry, P. rettgeri GFP) ? While being less quantitative, this could bring another set of data that support their main conclusions.On this line, the notion that the methods used here demonstrate bacterial growth in an individual fly (see language in Figure 3 legend) is grossly overstated. Clearly, as a population, the evidence clearly demonstrates bacterial growth in vivo. However, the bacterial growth within any individual fly cannot be assessed by these methods.

We agree with the reviewers that the analysis of the immune response is central to our argumentation. We limited ourselves to one time point only (8hrs), not because of experimental limitations, but because of theoretical assumptions. In our model, individual hosts behave similarly before the time to control, but after *t^c^* is reached by the population, individual flies show very different bacterial loads (depending on their level of control) and we know from other experiments that positive feedback amplifies the immune response in flies that have higher bacterial loads at later time points. Our assumption is therefore that before *t^c^*, the immune response results *only* from the detection of uncontrolled bacteria (could therefore correlate with the number of bacteria with high precision). After *t^c^*, additional processes occur, death of bacteria in some cases, continued growth of bacteria, accumulation of damage as a consequence of infection, and probably often a mix of all of the above. In very early time points, like 2hrs, the difference of bacterial load among individuals is so low that it would be difficult to detect a correlation. We therefore chose the latest possible time point that occurs before *t^c^* for our experiment.

In response to the suggestion of additional methods of monitoring AMP expression, we evaluated fluorescence from transgenic flies (DIG flies) bearing 4 copies of *Dpt-GFP*. Unfortunately, however, in both replicates of this experiment, the GFP signal was too low to quantify through the cuticle at 6-8 hours post-injection. At a later time point, 10-12hrs post-injection, we detected inter-individual variation in GFP signal. However, as stated above, this is after the point where some individuals have controlled their infections, while others did not (and will eventually never do). Thus, the variation may only reflect the individual variation in the number of bacteria, and would therefore not help demonstrate inter-individual variation in the initiation of the immune response upon infection.

We agree that we infer bacterial growth in individual flies from multiples snapshots of an infection, rather than following directly bacterial growth within a single fly. We have now mentioned that point in the Materials and methods (subsection “Model of growth dynamics”, third paragraph) and in the Results (subsection “A mixture model to capture within-host bacterial growth dynamics”, first paragraph).

3) Clarity of the text and relevant information:Many parameters affect infection such a temperature, sex, injection methods…. This information should be in the Results section not hidden in the Materials and methods. It is not clear what each survival graph shows, is it a representative result? How many times was it repeated? How many flies? Males or females? For bacterial load graphs, were all individual flies taken from one experiment (one biological replicate) or is it pooled from several replicates? It is often difficult to know how the experiments were done. All these details should be provided. Moreover, we could not exclude batch effect. This is actually suggested by Figure 4 with the new bacteria appearing more virulent.

We added more details describing the sample sizes and experimental structures in Materials and methods, Results and Figure legends. In general, our survival graphs are a representative example of the pool of replicated experiments but the analyses were performed taking vial and blocking effects into account. Our bacterial load experiments are also replicated, and often are additionally replicated in the context of another experiment. For instance, we quantified *Ecc15* load for Figure 3, and this was independently replicated to calculate *t^c^* in Figure 7. The same is true for the quantification of bacterial growth in an Imd mutant, which is a key result. The result is presented first in 4B but is replicated in 7A. We now mention in the manuscript that some results were confirmed later to point out this type of independent replication. In addition, the dual outcomes of infection we describe were replicated over several host genotypes. Altogether, we have replicated this phenomenon extensively, first for specific conditions, across genotypes, and generally across different bacteria.

Figure 4 represents a different type of experiment intended to test the possible effect of bacterial selection within the host as a mechanism for the variation in infection outcome. This experiment was not designed as other experiments in the manuscript. In this experiment, the bacteria injected were cultured directly from infected flies (of high or low bacterial burden) and subsequently injected to flies. This experiment demonstrates that passage through the host does not change the variation in infection outcome. We are very confident that the apparent increase in virulence is a consequence of how the bacteria were handled during this experiment specifically, and we have no evidence for evolved increase in virulence or experimental batch effects. Outside of this experiment, and in our infection settings, we note a robust replication of the outcomes of our infections.

4) All the data should be made publicly available to allow other scientists to use them. I would suggest to include all the raw data in excel sheets (bacterial count, survival data; biological repeats…).

All the raw data will be uploaded on Dryad.

5) The idea that the bacteria proliferate at the same rate in LB medium and flies should be documented for other strains.

We agree with the reviewer and have now added an experiment addressing this question specifically (subsection “A mixture model to capture within-host bacterial growth dynamics”, first paragraph and Figure 3—figure supplement 1). Specifically, we quantified the growth of additional bacterial species that we know are not controlled within the first 8 hours both in vitro and in vivo. Our results show that their growth in vitro did not differ significantly from their growth in Canton S flies measured in parallel. We now conclude that for multiple bacteria, the bacteria proliferate at the same rate in LB medium and flies early during infection before the *t^c^* occurs.

6) Description of the model: more information on the model should be included to the main text. As it is written it is hard to understand this section of the paper. Moreover, even as written it is not fully and clearly explained to a non-expert in the field of modeling. For example, the meaning and import of mini graph of "Frequency vs. t^c^ was unclear to one reviewer, and the relationship of P^c^ to the area below "the" curve was likewise unclear, as there are 2 curves presented. Some parameters are introduced but described later: V^c^ is shown in the figure but mentioned only in the Discussion. The confusion on the description of the model stem partly because of the high density at which series of hypotheses, tests and results are presented, and different sets of empirical results invoked (e.g., the third paragraph of the subsection “A mixture model to capture within-host bacterial growth dynamics”, which covers huge terrain very rapidly). Perhaps greater use of paragraph breaks or subheadings would help the reader navigate.Globally, a longer paragraph with the model, the definition of each parameter could improve the impact of this paper.

We have now added more information on the model including a new supplementary figure (Figure 6—figure supplement 1). This additional material can be found in the Materials and methods section (subsection “Model of growth dynamics”) and in the figure legends. In addition, a definition of the parameters can also be found in Table 1.

[Editors' note: further revisions were requested prior to acceptance, as described below.]

The manuscript has been improved but there are some remaining issues that need to be addressed before acceptance, as outlined below:Both the external reviewer and the reviewing editor found that the authors have adequately answered to most of reviewers’ comments. As you can see from the reviewer 3, there is regret that the paper does not include in vivo monitoring of antimicrobial peptide gene expression to back up the model. Nevertheless, it is true that GFP reporters are slow and might not be the appropriate tool to monitor the immune dynamic at early time point. To address this in particular, we would recommend that you adjust your text, in both the Results and the Discussion, to make it explicitly clear that you are inferring a variance in the initiation of response, but do not (yet) have tools available to demonstrate it.Reviewer #3:This remains one of the most thought provoking papers I have read in recent years. I think it will be a landmark in the field.That being said, I continue to struggle with the big issue, which is not directly addressed by the revisions. In short, the authors have inferred that the differences in the timing or strength of the initial immune response following infection determine the outcome (death at BLUD or chronic infection at SPBL). Surprisingly, they write in the rebuttal that they have been unable to measure AMP induction via fluorescent protein reporters at these early time points, at least in intact flies. Ideally, I would love some type of data that can probe this inference.

We share the disappointment of the reviewer as a non-invasive and quantitative method to measure the immune response in living flies would be invaluable. If we had such an assay, we could directly test the correlation of early immune induction with later bacterial load. Unfortunately, however, every sufficiently quantitative assay is destructive and every non-destructive assay is not sufficiently quantitative. Although GFP and dsRed transgenic reporters for immune system activity exist, the fluorescence is not strong enough in the first few hours of infection for reliable in vivoquantification by fluorescence microscopy through the cuticle. The signal becomes stronger much later in infection, but by then, a positive feedback loop between the number of bacteria and strength of immune activity has initiated resulting in a positive correlation between bacterial load and immune gene expression. Even in the best of circumstances, transgenic reporters provide indirect estimates of immune system activity. In addition, kinetics of GFP transcript and protein accumulation may be different from the endogenous AMPs. We therefore believe that RT-qPCR is the most precise method to quantify transcriptional expression of the immune response. To address our experimental question, we really would need to precisely and non-destructively measure activity of the immune system at very early stages of infection, before bacterial loads have begun to diverge among individuals, and such tools sadly do not exist right now.

Another more significant issue, is (still) the presentation of the mathematical model. Lots is made of the probability calculation in Figure 6 and beyond, but the way this probability is determine is not explained in the Results section. It is presented in detail in the Materials and methods, but frankly the math is over my head. I would prefer some discussion of this computation directly in the Results section so as to make sense of Figure 6, for example.

We understand the concern of the reviewer, and in order to explain how the probability is calculated, we added more detail to the Results (subsection “A mixture model to capture within-host bacterial growth dynamics”, first paragraph). The probability of control is computed as (1 – the probability that bacterial load reaches *n^tip^* before control is effective). However, the actual calculation is not simple and is not a primary result, so we prefer to keep it in the method section and supplement. We hope that sufficiently interested readers will probe the methods and supplement and we believe this strategy will minimize confusion for the more casual reader.

Finally, the rationale for the time points used in 7B (4 hour) and 7C (8 hours) is presented, and argued compellingly, in the rebuttal but this logic is still missing from the actual text of the article. I would recommend including.

We agree with the reviewer and added some explanation justifying the time points used in 7Cii (4hr) and 7D (8hr). (subsection “Variation in *T^c^* and inter-individual variation in infection outcome”, last paragraph).